# RAPID TRAINING OF HAMILTONIAN GRAPH NETWORKS USING RANDOM FEATURES

**Atamert Rahma**[1,2∗], **Chinmay Datar**[1,2,3], **Ana Čukarska**[1,2], **Felix Dietrich**[1,2,4]
[1]Technical University of Munich     [2]Munich Center for Machine Learning
[3]TUM Institute for Advanced Study     [4]Munich Data Science Institute

## ABSTRACT

Learning dynamical systems that respect physical symmetries and constraints remains a fundamental challenge in data-driven modeling. Integrating physical laws with graph neural networks facilitates principled modeling of complex N-body dynamics and yields accurate and permutation-invariant models. However, training graph neural networks with iterative, gradient-descent-based optimization algorithms (e.g., Adam, RMSProp, LBFGS) often leads to slow training, especially for large, complex systems. In comparison to 15 different optimizers, we demonstrate that Hamiltonian Graph Networks (HGN) can be trained 150-600× faster–but with comparable accuracy–by replacing iterative optimization with random feature-based parameter construction. We show robust performance in diverse simulations, including N-body mass-spring and molecular systems in up to 3 dimensions and 10,000 particles with different geometries, while retaining essential physical invariances with respect to permutation, rotation, and translation. Our proposed approach is benchmarked using a NeurIPS 2022 Datasets and Benchmarks Track publication to further demonstrate its versatility. We reveal that even when trained on minimal 8-node systems, the model can generalize in a zero-shot manner to systems as large as 4096 nodes without retraining. Our work challenges the dominance of iterative gradient-descent-based optimization algorithms for training neural network models for physical systems.

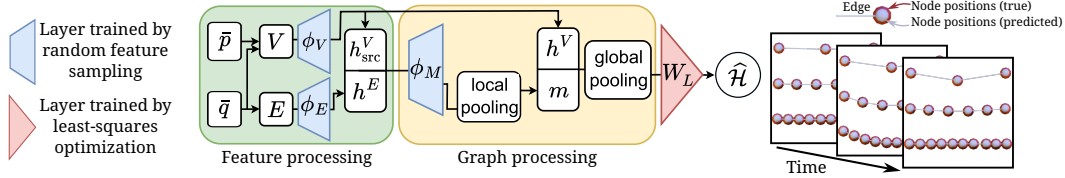

Figure 1: We propose an efficient training method for Hamiltonian graph networks using random feature sampling and linear solvers (left, also see Figure 3). The HGN captures ground truth dynamics of physical systems (shown: chain of 10 nodes, trained on 5) and trains up to 600× faster than State-Of-The-Art (SOTA) optimizers.

## 1 INTRODUCTION

**Learning from data** requires careful design in several key areas: the data, the model, and the training processes. To enable the model to generalize beyond the training set, it is important to incorporate a set of inductive biases into these processes. **When approximating physical systems**, it is beneficial to include physical priors to accurately capture the system's characteristics, including its dynamics and the fundamental physical laws (Tenenbaum et al., 2000; Chang et al., 2016; Watters et al., 2017; de Avila Belbute-Peres et al., 2018; Sharma & Fink, 2025). Consequently, many architectural designs are rooted in modeling physical frameworks, such as Hamiltonian mechanics (Bertalan et al., 2019; Greydanus et al., 2019), Lagrangian mechanics (Cranmer et al., 2019; Lutter et al., 2019;

---

∗Corresponding author, atamert.rahma@tum.de

Ober-Bloebaum & Offen, 2023), port-Hamiltonian systems (Desai et al., 2021; Roth et al., 2025), and GENERIC (Hernández et al., 2021; Lee et al., 2021; Zhang et al., 2022b; Gruber et al., 2025).

**Graph networks** have useful inductive biases such as locality and permutation invariance (Corso et al., 2024), which are desirable for many interconnected, complex systems observed in nature. Thus, for many applications in natural sciences, a graph network model is a suitable choice. The key aspects of modern graph networks include neural message passing (Gilmer et al., 2017) and encoding additional local information into the system (Corso et al., 2024; Schlichtkrull et al., 2018; Brockschmidt, 2020). In physics, graph networks have been employed to analyze data from the Large Hadron Collider (DeZoort et al., 2023), model mechanical systems (Zhao et al., 2024), and fluid dynamics (Xue et al., 2022; Peng et al., 2023; Li et al., 2024).

Efficient and robust training of graph networks on large systems for natural and life sciences is an active area of research. Despite the advantages of using graphs for physical N-body systems, their **training is reportedly slow due to gradient-descent-based iterative optimization** (Kose & Shen, 2023; Shukla et al., 2022; Vignac et al., 2020; Kumar et al., 2023; Zhao & Cheah, 2025; Marino et al., 2025). These challenges become even more pronounced when a numerical integrator is incorporated into the model architecture (Xiong et al., 2021). Furthermore, physics-informed models are often sensitive to the selection of hyperparameters (Shukla et al., 2022), which increases the challenges posed by slow iterative training.

Recently **random feature (RF) networks** have been shown to be promising for approximating physical systems (Fabiani et al., 2021; Datar et al., 2024; Rahma et al., 2024; Bolager et al., 2024; Fabiani et al., 2025; Galaris et al., 2022). However, to the best of our knowledge, random features have not been used to train graph networks for modeling physical systems. Recent work on RF-Hamiltonian neural networks (RF-HNNs) is promising (Rahma et al., 2024), where RF-HNNs are trained without using iterative algorithms. The authors demonstrate very low approximation errors, but only for very small systems and without leveraging the graph structure. In this paper, we introduce an efficient and accurate training method that utilizes random features for Hamiltonian Graph Networks (RF-HGNs, see Figure 1). Our main contributions are as follows.

- We **introduce Random Feature Hamiltonian Graph Networks**, combining random sampling with graph-based physics-informed models for the first time, and show how one can incorporate translation, rotation, and permutation invariance as well as knowledge about the physical system (see Section 3).

- We provide **a much faster and more efficient alternative to gradient-descent-based iterative optimization algorithms** for training that avoids challenges related to slow, non-convex optimization and vanishing or exploding gradients (see Section 4).

- We perform a **comprehensive optimizer comparison** with 15 different optimizers and demonstrate that random feature-based parameter construction offers up to 600 times faster training for HGNs, without sacrificing predictive performance (see Section 4.1). The demonstrations are performed on challenging benchmark problems, including mass-spring and molecular dynamics systems.

- We **demonstrate strong zero-shot generalization**, with models trained on graphs with tens of nodes accurately predicting dynamics on graphs with thousands of nodes (see Section 4.2).

## 2 RELATED WORK

**Training graph networks:** A graph structure allows for modeling a wide range of processes due to structural properties or underlying causal relationships. For many problems, the best way to achieve good performance is by using a large, high-quality dataset to train a large model. In such settings, training can be significantly slowed down due to the computational effort needed for backpropagation. Improvements can be achieved with specific sampling methods for the training data (Nagarajan & Raghunathan, 2023; Zhang et al., 2022a; Zhou et al., 2022; Kaler et al., 2022; Zhang et al., 2021; Lin et al., 2020), graph coarsening (Hashemi et al., 2024; Kumar et al., 2023; Jin et al., 2022a;b; Bravo Hermsdorff & Gunderson, 2019), or hardware acceleration(Shao et al., 2024; Gupta et al., 2024; Zhu et al., 2025; Wan et al., 2023; Yang et al., 2022; Kaler et al., 2022; Wang et al., 2021; Cai et al., 2021; Lin et al., 2020). Nevertheless, graph networks for physics still face challenges during

training due to a need for high accuracy in the dataset, irregular memory access, load imbalance during backpropagation (Shukla et al., 2022), and hyperparameter tuning (Schmidt et al., 2021).

**Graph networks for physics:**  A notable advantage of graph-based models is that they are tractable for high-dimensional data, assuming that the graph connectivity remains sparse, which is suitable for many physical systems learned in a data-driven way. Recent work has incorporated graph neural networks into their model architectures for approximating physical systems (Pfaff et al., 2021; Sanchez-Gonzalez et al., 2019; Sanchez-Gonzalez et al., 2020; Tierz et al., 2025; Varghese et al., 2025; Bhattoo et al., 2022; Thangamuthu et al., 2022). However, training a graph network for a very large number of nodes is challenging; one possible remedy is to partition a large graph and enable information exchange between partitions, which are trained individually (Nabian et al., 2024). Approaches for graph networks for Hamiltonian and Lagrangian systems are used by Thangamuthu et al. (2022); Bhattoo et al. (2022); Bishnoi et al. (2023); these models are typically trained with the Adam optimizer and applied to N-body systems. Other work addresses the issue of long-range information loss in large graphs by adding physics-based connections (Yu et al., 2025), yielding better predictions but not addressing training difficulties that might arise.

**Random features for graph networks:**  Random features originated with the idea of using a perceptron by Rosenblatt (1962) and gained traction after theoretical contributions established that they can lead to accurate approximations (Johnson & Lindenstrauss, 1984; Barron, 1993; Rahimi & Recht, 2007) at low computational cost. As the machine learning community gains a better understanding of random features (Rahimi & Recht, 2008; Bolager et al., 2023; Fabiani, 2024), many new variants are being explored (Zozoulenko et al., 2025; Bolager et al., 2024; Datar et al., 2024; Rahma et al., 2024). An innovative approach for graph classification problems used a random features approach and demonstrated competitive accuracy on large classification datasets with a training time of only a few seconds or minutes (Gallicchio & Micheli, 2020). Such an approach is related to echo state graph networks (Gallicchio & Micheli, 2010; Wang et al., 2023). Recent work has also developed graph random features enabling kernel methods on large graphs (Choromanski, 2023; Reid et al., 2023b;a), leading to a notable reduction of the cubic time-complexity for kernel learning.

## 3 METHOD

**Problem setup:**  In this study, we aim to efficiently learn the Hamiltonian of a dynamical system from observed phase space trajectories, while exploiting the underlying graph structure and incorporating relevant physical invariances into the model. We consider a target Hamiltonian for an N-body system on $\mathbb{R}^{2d \cdot N}$, the Euclidean phase-space of dimension $2d \cdot N \in \mathbb{N}$, where $d$ is the spatial dimension. We denote the generalized position and momentum vectors by $q, p \in \mathbb{R}^{d \cdot N}$, with $q_i, p_i \in \mathbb{R}^d$, denoting the $i^{\text{th}}$ particle's state. We denote by $\dot{x}$ the time derivatives of a trajectory $x(t) : \mathbb{R} \to \mathbb{R}^k$ for $k \in \mathbb{N}$. The Hamiltonian is a scalar-valued function $\mathcal{H} : \mathbb{R}^{2d \cdot N} \to \mathbb{R}$ that describes the system dynamics in the phase-space through Hamilton's equations (Hamilton, 1834; 1835) given by

$$\begin{bmatrix} \dot{q} \\ \dot{p} \end{bmatrix} = J \nabla \mathcal{H}(q, p), \quad J = \begin{bmatrix} 0 & I \\ -I & 0 \end{bmatrix} \in \mathbb{R}^{(2d \cdot N) \times (2d \cdot N)}, \tag{1}$$

where $I \in \mathbb{R}^{(d \cdot N) \times (d \cdot N)}$ is the identity matrix. We summarize the notation in Appendix A.

**Graph representation:** We focus on N-body systems in this work, where the graph representation is naturally available, e.g., a chain of masses connected via springs. Given $d_V, d_E \in \mathbb{N}$, we write the system with $N$ nodes as a graph $G = (V, E)$ with a node feature set $V = \{v_i \in \mathbb{R}^{d_V} \mid i = 1, \ldots, N\}$ and an edge feature set $E = \{e_{ij} \in \mathbb{R}^{d_E} \mid \forall i, j \text{ such that } A_{ij} = 1\}$, where $A \in \mathbb{R}^{N \times N}$ is the symmetric adjacency matrix that encodes the node connectivity information. We parametrize the Hamiltonian function $\mathcal{H}$ with a graph neural network, and then use the trained network to simulate the physical system by integrating Equation (1) with the symplectic Störmer-Verlet integrator ((Hairer et al., 2003), also see (Offen & Ober-Bloebaum, 2022)). In contrast to previous work on Hamiltonian Neural Networks (Bertalan et al., 2019; Greydanus et al., 2019; Dierkes et al., 2023), we train our networks through random feature sampling algorithms rather than iterative, gradient-descent-based optimization. Thus, we call our approach **"gradient-descent-free."**

## 3.1 ENCODING INVARIANCES

The systems we consider are translation-, permutation-, and rotation-invariant, i.e., when the whole system is shifted, permuted, or rotated, the Hamiltonian stays constant. With the general Hamiltonian formulation, however, we cannot guarantee such invariances by design (Du et al., 2022). To construct such invariant representations, we introduce transformed coordinates $\bar{q}, \bar{p} \in \mathbb{R}^{d \cdot N}$ derived from the original phase-space coordinates $q, p \in \mathbb{R}^{d \cdot N}$ defined in an arbitrary reference frame.

**Translation-invariant representation:** To make the position representation (and consequently the system representation) translation invariant, we normalize the positions by subtracting the mean $q_i \leftarrow q_i - \frac{1}{N} \sum_{i=1}^{N} q_i$. We do not make the generalized momenta $p$ translation invariant, as shifting the momenta would change the total energy of the system in N-body systems, for instance, when the kinetic energy depends on the norm of $p$.

**Permutation-invariance:** The graph structure and appropriate message passing algorithms inherently provide us with a system representation that is invariant with respect to node index permutation.

**Rotation-invariant representation:**
Starting from the translation-invariant representation, we then perform another transformation to make the final representation also rotation-invariant. Here, we explain how to encode a rotation-invariant representation for a single-body system $N = 1$ and spatial dimension $d = 2$ for brevity.

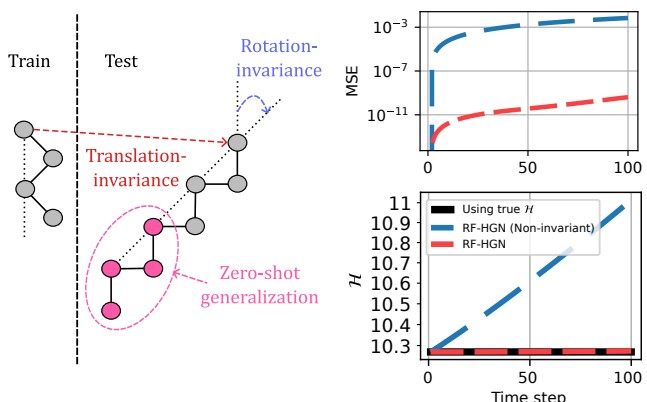

We construct a new representation in a local orthonormal basis starting from the original coordinates $p, q \in \mathbb{R}^2$. To construct the basis, we choose the first basis vector as $e_1 = \frac{q_1}{\|q_1\|} \in \mathbb{R}^2$, where $\|\cdot\|$ denotes the $l^2$ norm. We construct a second basis vector $e_2 = \mathcal{R}e_1 \in \mathbb{R}^2$, which is orthonormal to $q_1 \in \mathbb{R}^2$ and obtained by simply rotating $q_1$ by $90^o$ using a rotation matrix $\mathcal{R}$. We then define an orthonormal matrix with the two basis vectors as $\mathcal{B} = [e_1 \quad e_2]$. Finally, the rotation-invariant coordinates for any body $i \in \{1, \ldots, N\}$ are $\bar{q}_i = \mathcal{B}^\mathsf{T} q_i$.

Figure 2: Illustration of train and test N-body system positions showcasing the RF-HGN's translation- and rotation-invariance, and its zero-shot generalization capability, validated by conserved Hamiltonian and low trajectory prediction errors for the test data (see Appendix B.1 for details).

Given a fixed first point, our procedure yields a rotation-invariant representation. One can uniquely identify the first point, independent of node ordering or orientation, as the one closest to the mean $\bar{q}$. In case of ties, we select the point with the smallest angle relative to the first coordinate axis centered at $\bar{q}$. The same procedure is applied to obtain rotation-invariant representations of the momenta. In higher dimensions, when $d > 2$ and $N > 1$, we follow a similar procedure, but construct the orthogonal bases using Gram-Schmidt orthogonalization instead (see Appendix B.2). Figure 2 demonstrates how translating the N-body system, rotating it, and even adding new nodes without re-training (zero-shot-generalization), yields low trajectory errors while conserving the Hamiltonian.

## 3.2 MODEL

We now describe each component of the architecture of the Hamiltonian Graph Network (HGN) in detail (see Figure 3). Please refer to Appendix B.3 for details on the forward pass.

### 3.2.1 NODE AND EDGE ENCODING

**Node features:** For an N-body system with translation- and rotation-invariant position and momenta representations, we define node features as $v_i = \begin{bmatrix} \bar{q}_i^\mathsf{T} & \bar{p}_i^\mathsf{T} \end{bmatrix}^\mathsf{T} \in \mathbb{R}^{d_V}$, where $d_V = 2d$, for each $i \in \{1, \ldots, N\}$. We define the set $V := \{v_i \,|\, i = 1, \ldots, N\}$ that collects all node encodings.

Figure 3: Random-feature Hamiltonian graph neural network architecture. **Left (green box):** Construction of node and edge encodings $h_{src}^V$ and $h^E$ from translation and rotation invariant position $q$ and momenta $p$ representations of an N-body system. **Right (orange box):** Construction of a global encoding for the graph using message passing. In RF-HGN, dense layers (blue) are constructed with random features, and linear layer weights (red) are optimized by solving a linear problem.

**Edge features:** We define the edge features for each edge $(i,j)$ with $i > j$ as $e_{ij} = \left[(\bar{q}_i - \bar{q}_j)^\mathsf{T}; \|\bar{q}_i - \bar{q}_j\|\right]^\mathsf{T} \in \mathbb{R}^{d_E}$, where $d_E = d + 1$ and take the absolute value $|(\bar{q}_i - \bar{q}_j)^\mathsf{T}|$ in the molecular system examples. We collect all the edge feature encodings in the set $E := \{e_{ij} \mid i > j \text{ and } i, j \in \{1, \dots, N\}\}$. In order to reduce the memory and computation costs, we define a canonical direction by always computing edge features from higher to lower-indexed nodes, such that each edge is represented only once and set $(e_{ji} = e_{ij})$. We use the relative displacement vector $\bar{q}_i - \bar{q}_j$ and its norm to represent the direction and distance between connected nodes in the system, in order to capture local geometric structure and pairwise interaction properties.

**Input encoding:** The constructed node and edge features are then encoded via separate dense layers,

$$h_i^V = \phi_V(v_i) = \sigma(W_V\, v_i + b_V) \in \mathbb{R}^{d_h} \quad \forall v_i \in V, \text{ and} \tag{2}$$

$$h_{ij}^E = \phi_E(e_{ij}) = \sigma(W_E\, e_{ij} + b_E) \in \mathbb{R}^{d_h} \quad \forall e_{ij} \in E, \tag{3}$$

where $\phi_V : \mathbb{R}^{d_V} \to \mathbb{R}^{d_h}$, and $\phi_E : \mathbb{R}^{d_E} \to \mathbb{R}^{d_h}$ are outputs of dense layers that encode the node and edge features, respectively, with weights $W_V \in \mathbb{R}^{d_h \times d_V}, W_E \in \mathbb{R}^{d_h \times d_E}$ and biases $b_V, b_E \in \mathbb{R}^{d_h}$. We denote the activation function (here, `softplus` or `gelu`) by $\sigma$. Using the symmetric edge features described earlier avoids duplicate memory and computation overhead in the input encoding as well, since we only compute the encoding $h_{ij}^E$ of each undirected edge feature $e_{ij}$ where $i > j$, and use the same encoding for both directions $h_{ij}^E$ and $h_{ji}^E$.

### 3.2.2 MESSAGE PASSING AND FINAL LAYER

We perform bidirectional message passing along edges $(i,j)$, where $A_{ij} = 1$, allowing nodes to aggregate information from their local neighborhoods.

**Message construction:** Messages are constructed from the encodings of source and edge nodes via a dense layer $\phi_M : \mathbb{R}^{2d_h} \to \mathbb{R}^{d_M}$ as

$$h_{ij}^M = \phi_M\left(\begin{bmatrix} h_i^V \\ h_{ij}^E \end{bmatrix}\right) = \sigma\left(W_M \begin{bmatrix} h_i^V \\ h_{ij}^E \end{bmatrix} + b_M\right) \in \mathbb{R}^{d_M}, \tag{4}$$

with weights $W_M \in \mathbb{R}^{d_M \times \mathbb{R}^{2d_h}}$ and biases $b_M \in \mathbb{R}^{d_M}$.

**Message passing (local pooling):** Each node aggregates incoming messages using a permutation-invariant operation (here, summation) $m_j = \sum_{i \in \mathcal{N}_j} h_{ij}^M$, where $\mathcal{N}_j$ is the set of neighbors of node $j$ (source of incoming edges to $j$ where $A_{ij} = 1$).

**Graph-level representation (global pooling):** All node embeddings and aggregated messages are pooled to form a global encoding of the network, such that

$$h_G = \sum_{j=1}^N \begin{bmatrix} h_j^V \\ m_j \end{bmatrix} \in \mathbb{R}^{d_L}, \text{ where } d_L = d_h + d_M. \tag{5}$$

**Linear layer:** The graph representation is linearly mapped to a scalar value that approximates the conserved value (energy) of the system, such that $\widehat{\mathcal{H}} = W_L \cdot h_G + b_L$, where $W_L \in \mathbb{R}^{d_L}$ and $b_L \in \mathbb{R}$ denote weights and bias of the linear layer, respectively. Without loss of generality, we omit the bias

term, as it acts only as an integration constant and does not affect the dynamics $\frac{\partial \mathcal{H}}{\partial q}$ and $\frac{\partial \mathcal{H}}{\partial p}$, which are our primary interest. We assume this constant is known to align the model's conserved quantity with the true Hamiltonian $\mathcal{H}$ in all examples.

## 3.3 TRAINING

We now describe the central idea of this paper – our training algorithm, where we use random feature sampling techniques instead of gradient-descent-based, iterative optimization algorithms. In particular, we discuss how to compute dense and linear layer parameters of the network.

**Dense layer parameters:** We compute the weights and biases $W_V, W_E, W_M, b_V, b_E, b_M$ of all the dense layers ($\phi_V, \phi_E, \phi_M$) using random sampling algorithms. Specifically, we use two sampling approaches here: Extreme Learning Machines (ELM) (Schmidt et al., 1992; Pao & Takefuji, 1992; Huang et al., 2004; 2006; Rahimi & Recht, 2008; Zhang et al., 2012; Leung et al., 2019) and the "Sample Where It Matters" (SWIM) algorithm (see (Bolager et al., 2023)) for unsupervised learning problems (see (Rahma et al., 2024; Datar et al., 2024)). As an illustrative example, we describe how to compute the parameters of the dense layer $\phi_V$ and use the notation from Equation (2). The parameters of the other dense layers are sampled analogously.

The **(ELM) RF-HGN** approach is data-agnostic. The weights $W_V$ are sampled from the standard normal distribution, and biases $b_V$ from the standard uniform distribution. The **(SWIM) RF-HGN** approach is data-driven. The network parameters are computed from pairs selected uniformly at random from input data points $x_i$, where the point coordinates correspond to dense layer inputs. The weight and the bias of the $i^{\text{th}}$ neuron in the dense layer are constructed using the input data pair $(x_i^{(1)}, x_i^{(2)})$ chosen uniformly at random from all possible pairs, so that $w_i = s_1(x_i^{(2)} - x_i^{(1)})\|x_i^{(2)} - x_i^{(1)}\|^{-2}$ and $b_i = -\langle w_i, x_i^{(1)} \rangle - s_2$. Here, $(w_i, b_i)$ are the weight and bias of the $i^{\text{th}}$ neuron, and $(s_1, s_2)$ are constants depending on the activation function used in the dense layer (see (Bolager et al., 2023) for an analysis).

**Linear layer parameters:** After sampling all the dense layer parameters, we compute the optimal parameters for the linear output layer of the network by computing the least squares solution (see (Rahma et al., 2024), but also related work (Bertalan et al., 2019)). For an N-body system, we denote a single input to the RF-HGN by $y \in \mathbb{R}^{2d \cdot N}$ and the output of the global pooling layer by $\Phi(y) \in \mathbb{R}^{d_L}$, and the total number of input data points by $M$. The linear system that approximately satisfies Hamilton's equations is then

$$\underbrace{\begin{bmatrix} \nabla\Phi(y_1) & \cdots & \nabla\Phi(y_M) & \Phi(y_0) \\ 0 & \cdots & 0 & 1 \end{bmatrix}^{\mathsf{T}}}_{Z \in \mathbb{R}^{(2d \cdot N \cdot M + 1) \times (d_L + 1)}} \cdot \underbrace{\begin{bmatrix} W_L^{\mathsf{T}} \\ b_L \end{bmatrix}}_{\theta_L \in \mathbb{R}^{d_L + 1}} \stackrel{!}{=} \underbrace{\begin{bmatrix} J^{-1}\dot{y}_1 & \cdots & J^{-1}\dot{y}_M & \mathcal{H}(y_0) \end{bmatrix}^{\mathsf{T}}}_{u \in \mathbb{R}^{2d \cdot N \cdot M + 1}}. \quad (6)$$

Equation (6) is solved for the linear layer parameters $W_L$ and $b_L$ using $l^2$ regularization. The regularization constant in most examples was chosen very small (see Appendix D). We assume the true Hamiltonian value $\mathcal{H}(y_0)$ to be known for a single data point to fix the integration constant $b_L$. We assume there is no external force acting on the system during training, such that the total energy is conserved. However, we can easily add an external force while evaluating the trajectory during inference. In the computational experiments, we mostly train with explicitly given time derivatives $\dot{x}$. We demonstrate training the model purely from time series data as part of the benchmark experiments (Section 4.4). Equation (6) results in a convex optimization problem, $[W_{L+1}^{\mathsf{T}} b_{L+1}]^{\mathsf{T}} = \arg\min_{\theta_L} \|Z\theta_L - u\|^2$, which can be solved using efficient least-squares algorithms (Meng et al., 2014b).

**Runtime and memory complexity:** During training, the sampling of dense layer parameters and gradient computation are fast, with the primary run-time bottleneck being the least squares solve (Equation (6)). Assuming $d_L \ll K = 2d \cdot N \cdot M$ (which is always the case in our experiments), the total run-time complexity is $\mathcal{O}(Kd_L^2)$. This highlights an important feature of our approach: training time scales linearly with data size $M$, the number of particles $N$, and the spatial dimension $d$, given fixed settings for other variables. The memory complexity during training is $\mathcal{O}(MN_e)$ and thus also scales linearly with the number of edges, and dataset size (see (Bolager et al., 2023) and Appendix B.4 for details).

## 4 COMPUTATIONAL EXPERIMENTS

We evaluate our method on mass-spring and molecular dynamics systems with two and three degrees of freedom (2D and 3D), as illustrated in Figure 4. We provide further details on the used datasets, setup, and hardware in Appendix C, Appendix D, and Appendix E, respectively. We additionally discuss hyperparameter tuning and ablation studies of increasing feature widths and number of message passes in Appendix F, test robustness against noise in Appendix H.4, demonstrate batch-wise training in Appendix H.5, and provide further random feature benchmarks in Appendix H.6. For $x_{\text{true}}, x_{\text{pred}} \in \mathbb{R}^m$ for $m \in \mathbb{N}$, we define the relative error as $||x_{\text{true}} - x_{\text{pred}}||_2/||x_{\text{true}}||_2$.

### 4.1 BENCHMARKING AGAINST SOTA OPTIMIZERS

Table 1: Results of training the HGN architecture for the 3D lattice system (see Figure 4 (a)) with different optimizers. Results show **mean (min, max)** over three runs on the same GPU hardware.

| Optimizer | Test MSE | Train time [s] | Speed-up |
|---|---|---|---|
| **RF-HGN (ours)** | 8.95e-5 (6.96e-5, 1.13e-4) | **0.16 (0.13, 0.22)** | - |
| LBFGS (Liu & Nocedal, 1989) | **3.56e-5 (1.21e-5, 7.94e-5)** | 23.85 (23.71, 23.95) | 148.96× |
| Rprop (Riedmiller & Braun, 1993) | 9.59e-4 (7.49e-5, 2.63e-3) | 30.84 (30.74, 30.94) | 192.62× |
| RMSprop (Tieleman & Hinton, 2012) | 1.09e-3 (2.55e-5, 3.18e-3) | 91.62 (91.13, 92.42) | 572.24× |
| Adam (Kingma & Ba, 2015) | 2.90e-3 (4.11e-5, 8.53e-3) | 91.64 (89.97, 92.66) | 572.37× |
| SGD+momentum (Sutskever et al., 2013) | 4.23e-3 (3.81e-3, 4.66e-3) | 91.65 (91.14, 92.20) | 572.43× |
| SGD (Robbins, 1951) | 2.36e-2 (1.85e-2, 2.86e-2) | 91.75 (91.51, 91.91) | 573.07× |
| Adagrad (Duchi et al., 2011) | 2.58e-2 (2.88e-3, 7.05e-2) | 92.03 (91.58, 92.47) | 574.84× |
| AdamW (Loshchilov & Hutter, 2019) | 2.91e-3 (4.30e-5, 8.53e-3) | 92.15 (91.86, 92.31) | 575.59× |
| Adamax (Kingma & Ba, 2015) | 1.85e-3 (1.55e-4, 4.32e-3) | 92.33 (92.07, 92.69) | 576.68× |
| Adadelta (Zeiler, 2012) | 8.11e-3 (1.49e-3, 1.96e-2) | 92.60 (92.25, 93.07) | 578.39× |
| Radam (Liu et al., 2021) | 1.69e-3 (5.36e-5, 4.75e-3) | 93.00 (92.75, 93.42) | 580.88× |
| Nadam (Dozat, 2016) | 9.11e-4 (4.08e-5, 2.61e-3) | 93.42 (92.88, 93.80) | 583.54× |
| Averaged SGD (Gower et al., 2019) | 2.36e-2 (1.85e-2, 2.86e-2) | 94.50 (94.01, 94.78) | 590.26× |
| Adafactor (Shazeer & Stern, 2018) | 2.41e-3 (1.06e-3, 4.71e-3) | 96.36 (95.67, 96.88) | 601.92× |

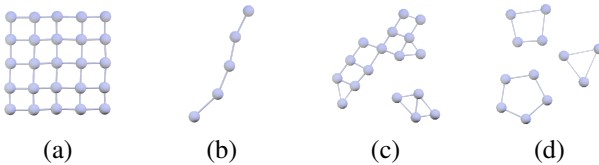

|      |      |      |      |
|:----:|:----:|:----:|:----:|
| (a) | (b) | (c) | (d) |

Figure 4: Graphs considered in the experiments: **(a)** 3D lattice (nodes arranged on a 2D grid, moving in a 3D space - see Section 4.1 and Section 4.2), **(b)** an open chain (nodes moving in 2D space - see Section 4.2 and 4.3), **(c)** molecules interacting through Lennard-Jones potential (nodes moving in 2D space with dynamic edges - see section Section 4.2), and **(d)** 2D closed chain (nodes moving in 2D space - see Section 4.4).

The goal of this experiment is to demonstrate the efficiency of our training approach in comparison with the conventional training methods that rely on SOTA iterative optimization algorithms. To this end, we consider all of the existing optimizers available in PyTorch (Paszke et al., 2019) as the current SOTA iterative training procedures. In this experiment, the target function is the Hamiltonian of a generalized $N_x \times N_y$-body lattice mass-spring system with a spatial dimension $d = 3$ given by Equation (C.8). Table 1 lists the results of training the HGN architecture for the lattice system with different optimizers, sorted by training time. The hyperparameters are tuned for each optimizer separately, and early stopping was used for all iterative approaches. **Our proposed training method significantly outperforms all iterative approaches in terms of training time** by a factor of **148 up to 601**, and is only slightly less accurate compared to the LBFGS method (a second-order optimizer). Section 4.4 includes comparisons for other graph network architectures on benchmark datasets, where we observed similar results.

### 4.2 ZERO-SHOT GENERALIZATION AND COMPARISON OF RANDOM FEATURE METHODS

We now study zero-shot generalization, where we train an RF-HGN on small systems of size 2x2, 3x3, and 4x4, and test on systems going from 2x2 up to 100x100. Figure 5 shows that we can accurately approximate a Hamiltonian for much smaller systems with 3x3 nodes and reliably predict it with extremely large systems of size 100x100 without retraining. A 2x2 system is an edge case where all

nodes have only two edges, lacking the nodes with four edges, as in the test data, explaining the poor zero-shot generalization.

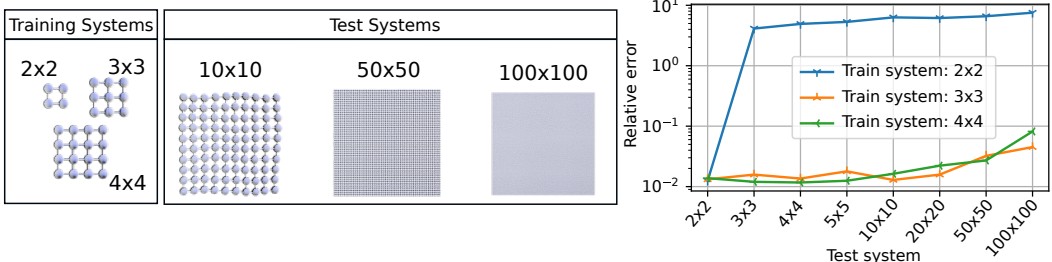

Figure 5: Illustration of accurate zero-shot generalization for 3D lattice (see Figure 4 (a)): Training on smaller systems (left) enables accurate predictions (right) on extremely large test systems (middle).

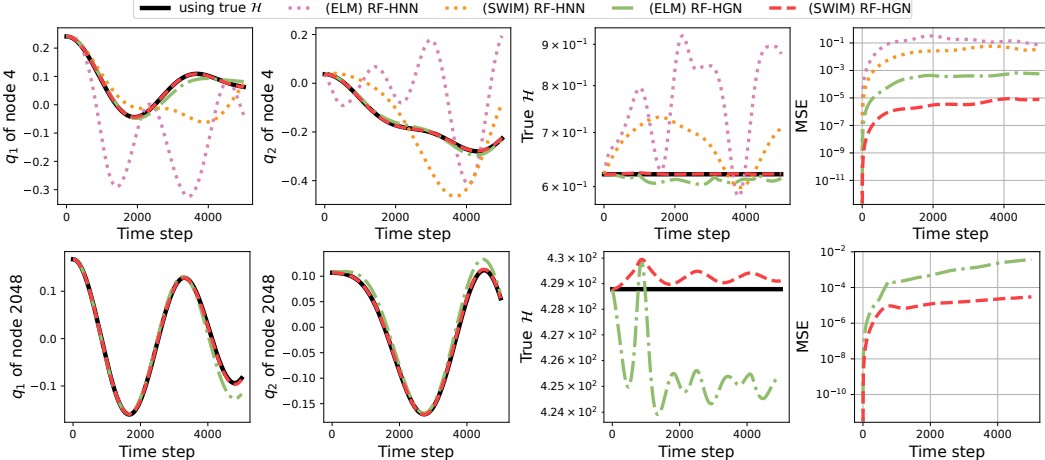

Figure 6: Illustration of position trajectories (first two columns), their true Hamiltonian values (third column), and MSE (fourth column) over time from models trained on a system with eight nodes on the 2D open chain (see Figure 4 (b)). Top row: Results from RF-HNN and RF-HGN architectures are visualized along with the ground truth. A system of the same size for training and testing is used ($2^3$ nodes). Bottom row: Results from RF-HGN architectures and ground truth for a zero-shot test case with a system size of $2^{12}$ nodes (trained on $2^3$ nodes).

We consider another example of an N-body chain mass-spring system in a 2D space (eq. (C.9)). Figure H.16 shows that by training on a much smaller system with $2^3$ nodes, RF-HGN trained with ELM and SWIM results in very low errors for systems as large as $2^{12}$ nodes. This demonstrates robust and strong zero-shot generalization in graph-based architectures, with training using SWIM outperforming ELM by approximately an order of magnitude. Since HNNs are not graph-based architectures, they have to be re-trained for each system, and even after re-training, they perform much poorly in comparison with graph-based architectures (by 1-2 orders of magnitude). We also observe that the graph-based architectures (RF-HGN) are slower to train compared to their respective counterparts trained with fully connected networks (RF-HNN), but are more accurate by 1-2 orders of magnitude, especially for large systems, even without re-training.

Figure 6 (top row) shows that the trajectories evaluated with the RF-HGN accurately match the true ones closely while approximately conserving the Hamiltonian, unlike RF-HNN. The RF-HGN trained with SWIM, in particular, outperforms the one trained with ELM by roughly two orders of magnitude. In the bottom row of Figure 6, we show zero-shot results, which are limited to only the graph networks. Our (SWIM) RF-HGN is robust and again exhibits a low error, while the ELM-trained model exhibits deviations from the true Hamiltonian.

We now consider another Hamiltonian, using the Lennard-Jones potential (eq. (C.10)) to investigate generalization properties to different geometries. First, a small system with 9 particles was trained

with Adam and the random feature methods ELM and SWIM. The accuracy with ELM was poor (see Figure H.18), and thus the results are omitted in Figure H.19. We speculate that the main reason for poor performance is the non-isotropic input variables, which makes normal distributions a bad choice for the weights. The results are given in Table 2, Figure H.19, Table H.32, Table H.33, and Figure H.18. To evaluate RF-HGN in a more complex scenario, we employed dynamic edge indices with a cutoff of 2.0, trained our model with 36 particles, and tested with 64 particles to test zero-shot generalization. We visualize the rollout trajectories in Figure H.20 and Figure H.21, and observe that SWIM sampling clearly outperformed ELM, while maintaining slightly worse approximation than the Adam-trained HGN. We note that none of the trainers could reach low approximation errors ($\sim 10\%$ relative error). To the best of our knowledge, our RF-HGN is the first random feature-based physics-informed graph network, and can be trained approximately 100 times faster than with the Adam optimizer at comparable accuracy.

Table 2: Molecular dynamics evaluation with 9 particles. Mean squared error (MSE) and relative $l^2$ error (rel. $l^2$) are reported together with the true Hamiltonian over the ground-truth trajectory and the (SWIM) RF-HGN predicted quantity over the rolled-out trajectory.

|  | T=1 | T=25000 | T=50000 | T=74999 | T=99999 |
|---|---|---|---|---|---|
| $q$ MSE | 1.140e-13 | 3.239e-03 | 1.998e-02 | 4.932e-02 | 8.301e-02 |
| $q$ rel. $l^2$ | 2.346e-07 | 3.978e-02 | 9.717e-02 | 1.545e-01 | 2.032e-01 |
| True $\mathcal{H}$ | -1.233e+01 | -1.233e+01 | -1.233e+01 | -1.233e+01 | -1.233e+01 |
| Model $\widehat{\mathcal{H}}$ | -1.223e+01 | -1.223e+01 | -1.248e+01 | -1.214e+01 | -1.217e+01 |

### 4.3 BENCHMARKING WITH REAL-WORLD POTENTIALS WITH INCREASING COMPLEXITY

To further evaluate the applicability of RF-HGN, we experiment with additional potentials from quantum mechanics (anharmonic oscillator Atkins et al. (2023) and molecular dynamics (the Morse potential Morse (1929)), and list testing results with an unseen initial condition in Table 3. In addition to using more complicated potential, we also now apply an external (gravitational) force node-wise during integration and simulate for a long-time horizon (see Figure H.10, H.12, H.14 for learned Hamiltonian plots over the trajectory; see Figure H.11, H.13, H.15 snapshot visualization of the predicted trajectories). The results show that more challenging potentials (non-linear forces) than the standard mass-spring (linear force) can also be approximated with the RF-HGN model with reasonable accuracy, compared to the Adam optimizer, still achieving 200-300× speed-ups, even without GPU acceleration. See Table 3 for details, and note that the RF-HGN results are achieved without extensive hyperparameter tuning (as opposed to Adam).

Table 3: Zero-shot (trained with 5 nodes, tested with 10 nodes) test evaluation by solving an unseen initial condition of a 2D chain (see Figure 4 (b)) using Hamiltonian graph models trained differently. The system is solved for 10,000 time steps with time step size $\Delta t = 10^{-2}$ with gravitational force applied during integration, and the last position MSE is reported against the reference solution using the true Hamiltonian. The results are listed for the standard spring, anharmonic oscillator, and the Morse potential in that order.

| Potential | (Adam) HGN | (ELM) RF-HGN | (SWIM) RF-HGN |
|---|---|---|---|
| $V(r) = \frac{1}{2}\beta r^2$ | 3.875e-03 | 2.331e-03 | 3.408e-05 |
| $V(r) = \frac{1}{2}\beta r^2 + \frac{1}{4}\eta r^4$ | 4.562e-02 | 4.324e-02 | 5.232e-04 |
| $V(r) = D(1 - \exp(-ar))^2$ | 8.893e-02 | 7.398e-04 | 1.218e-03 |

### 4.4 BENCHMARKING AGAINST SOTA ARCHITECTURES

The goal of this benchmark is to compare the results of our model with the existing state-of-the-art graph-based network architectures used to model physical systems. The Adam optimizer (Kingma & Ba, 2015), widely regarded as the SOTA optimizer for physics-informed GNNs (Kumar et al., 2023; Thangamuthu et al., 2022), is used as the default in our comparisons. We use the dataset and code from Thangamuthu et al. (2022), introduced at the NeurIPS 2022 Datasets and Benchmarks Track. In line with the other experiments, we observe excellent performance on the benchmark spring systems (Figure 4, d) with orders of magnitude faster training times while maintaining a comparable accuracy. Training times are reported in Table 4. In our evaluations, certain specialized architectures such as

Lagrangian Graph Networks (LGN) (Bhattoo et al., 2022) occasionally exhibit instability and diverge on test trajectories, whereas our model maintains robust performance. For a detailed problem setup, accuracy comparison, and the architectures, please see Appendix G.

Table 4: Comparison of training times (in seconds) for RF-HGN optimized using SWIM to existing physics-informed graph models optimized with Adam on a benchmark dataset from Thangamuthu et al. (2022) on a 2D closed chain (see Figure 4 (d)).

| System size | (SWIM) RF-HGN | FGNN | FGNODE | GNODE | LGN | LGNN | HGN | HGNN |
|---|---|---|---|---|---|---|---|---|
| $N = 3$ | **2.51** | 406.14 | 380.35 | 2367.37 | 12534.81 | 7225.88 | 1288.08 | 3568.12 |
| $N = 4$ | **3.87** | 475.24 | 430.32 | 2499.04 | 20536.78 | 6259.58 | 1370.14 | 4021.59 |
| $N = 5$ | **5.42** | 536.27 | 520.54 | 2600.31 | 53148.24 | 8774.59 | 1676.78 | 4380.46 |

## 5 CONCLUSION

We propose a training algorithm for Hamiltonian graph networks via rapid random feature sampling and linear system solvers. Our approach completely avoids slow, iterative gradient-descent-based optimization, which is especially challenging in the graph network and the physics-informed settings. We demonstrate our approach on chain, lattice, and molecular systems in up to three spatial dimensions, encompassing N-body systems. By incorporating translation, rotation, and index-permutation invariances, we extend random feature methods to graph-based Hamiltonian network architectures. Compared to 15 optimizer baselines, our method offers dramatic speedups (100× to 1000×) while achieving competitive accuracy in 3D physical systems. Remarkably, training on $3 \times 3$ systems suffices to accurately predict dynamics in systems of size $100 \times 100$, demonstrating strong zero-shot generalization capabilities. With this generalization from such small-scale training systems, one can deploy models without needing to re-train on full-scale data, enabling fast prototyping.

**Limitations and future work:** For very small graphs, the HNN architecture is often faster to train than a graph-based approach, making it a better choice than HGN in these cases. Generalization capabilities of the HGN models are typically limited to the same type of graphs, i.e., models trained on chains (edge degrees up to two) cannot be used to predict dynamics of lattices (edge degrees up to four) (Corso et al., 2024). We observed similar challenges when using dynamic edges in the molecular dynamics examples for all optimizers. Therefore, for those examples, one could either use more data or a less physically complex model (e.g., an SE(3) graph network that provides SE(3) equivariance by construction (Du et al., 2022)) to reduce data requirements. Our approach does not easily generalize to other graph neural network architectures (e.g., convolution or self-attention on the individual node features). In future work, we intend to extend this work by employing multiple message passing layers and deeper architectures where random feature boosting might help (Zozoulenko et al., 2025).

**Ethics statement:** We demonstrate that data-driven construction of random features can significantly outperform many SOTA optimizers in terms of accuracy and training time. The tremendously increased training speeds we report may also speed up the development of nefarious and even dangerous applications. Similar to all HGN and HNN models, our specific training method is not designed for this purpose. However, specific bad intent as well as significant further development would be required for this to happen. We hope that, instead, our work has a profound positive societal impact in the future, because training such accurate models from data is important in many sciences as well as in engineering – but has been slow up to now, due to the difficulties in training.

**Reproducibility statement:** We provide details on the used datasets, model setup with hyperparameters, and hardware details in Appendix C, Appendix D, and Appendix E, respectively for all the numerical experiments discussed in the main text and in the appendix. In the supplementary materials we provide further instructions in a `README.md` on how to reproduce all results in the form of tables and plots. Our codebase is publicly available at

$$\texttt{https://gitlab.com/fd-research/swimhgn.}$$

**Acknowledgments:** We are grateful for discussions with Samuel James Newcome, Manish Kumar Mishra, Markus Mühlhäußer, Jonas Schuhmacher, Iryna Burak, Nadiia Derevianko, Qing Sun, and Erik Lien Bolager. A.R. is supported by the BMBF (project AutoMD-AI), A.Č. is supported by the TUM Georg Nemetschek Institute – Artificial Intelligence for the Built World., and C.D. and F.D. are supported by the DFG (project no. 468830823), and acknowledge association with DFG-SPP-22. C.D. is partially funded by the Institute for Advanced Study (IAS) at the Technical University of Munich. The authors gratefully acknowledge the computational and data resources as well as the support provided by the Leibniz Supercomputing Centre (www.lrz.de).

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

Table A.5: Notation used in Section 3 and in the appendix.

| **Problem setup** | |
|---|---|
| $N$ | Number of nodes/particles in the system |
| $d$ | Spatial dimension |
| $M$ | Number of training data points |
| $N_e$ | Number of edges in the graph |
| $q, p \in \mathbb{R}^{d \cdot N}$ | Positions and momenta of all the nodes |
| $q_i, p_i \in \mathbb{R}^d$ | Positions and momenta of the $i^{\text{th}}$ node |
| $\mathcal{H} : \mathbb{R}^{2d \cdot N} \to \mathbb{R}$ | True Hamiltonian function |
| $\widehat{\mathcal{H}} : \mathbb{R}^{2d \cdot N} \to \mathbb{R}$ | Predicted Hamiltonian function |
| $A$ | Symmetric adjacency matrix |
| **RF-HGN setup** | |
| $V$ | Set of node feature encodings |
| $E$ | Set of edge feature encodings |
| $d_V \in \mathbb{N}$ | Number of node features |
| $d_E \in \mathbb{N}$ | Number of edge features |
| $d_h \in \mathbb{N}$ | Latent (hidden) dimension for encoding node and edge features |
| $d_M \in \mathbb{N}$ | Latent (hidden) dimension for encoding messages |
| $d_L \in \mathbb{N}$ | Size of the input to the linear layer (here also the network width) |
| $\bar{q}_i, \bar{p}_i \in \mathbb{R}^d$ | Invariant representation |
| $v_i \in \mathbb{R}^{d_V}$ | Node features for the $i^{\text{th}}$ node |
| $e_{ij} \in \mathbb{R}^{d_E}$ | Edge features for the edge $(i, j)$ with $i > j$ |
| $\mathcal{N}_j$ | Set of source nodes connected to destination node $j$ |
| $\phi_V : \mathbb{R}^{d_V} \to \mathbb{R}^{d_h}$ | dense layer for encoding the node features |
| $\phi_E : \mathbb{R}^{d_E} \to \mathbb{R}^{d_h}$ | dense layer for encoding the edge features |
| $\phi_M : \mathbb{R}^{2d_h} \to \mathbb{R}^{d_M}$ | Message encoder |
| $W_V \in \mathbb{R}^{d_h \times d_V}, b_V \in \mathbb{R}^{d_h}$ | Weights and biases of the node encoder |
| $W_E \in \mathbb{R}^{d_h \times d_E}, b_E \in \mathbb{R}^{d_h}$ | Weights and biases of the edge encoder |
| $W_M \in \mathbb{R}^{d_M \times \mathbb{R}^{2d_h}}, b_M \in \mathbb{R}^{d_M}$ | Weights and biases of the message encoder |
| $W_L \in \mathbb{R}^{d_L}, b_L \in \mathbb{R}$ | Weights and biases of the linear layer |
| $m_j \in \mathbb{R}^{d_M}$ | aggregated incoming message of node $j$ |
| $h_i^V \in \mathbb{R}^{d_h}$ | Node feature encoding for the $i^{\text{th}}$ node |
| $h_{ij}^E \in \mathbb{R}^{d_h}$ | Edge feature encoding for the edge defined by nodes $i, j$ |
| $h_{ij}^M \in \mathbb{R}^{d_M}$ | Constructed message encoding |
| $h_G \in \mathbb{R}^{d_L}$ | Global encoding of the graph |

APPENDIX

## A  NOTATION

We define the notation we use in the main text and in the Appendix in Table A.5.

## B  ADDITIONAL DETAILS ON RANDOM-FEATURE HAMILTONIAN GRAPH NETWORKS

We discuss the problem setup used in Figure 2 in Appendix B.1, how to construct rotation-invariant representation for spatial dimension $d = 3$ in Appendix B.2, the algorithm for the forward pass for RF-HGN in Appendix B.3 and run-time and memory complexity in Appendix B.4.

### B.1  ADDITIONAL DETAILS FOR FIGURE 2

The initial conditions for training the networks in Figure 2 are generated by displacing the positions by $dq \sim U(-0.5, +0.5)$ and with momenta $p \sim U(-2, +2)$ from some fixed reference frame illustrated

on the left of Figure 2. The system is integrated using symplectic Störmer-Verlet Hairer et al. (2003) with $\Delta t = 10^{-4}$ for 100 steps using the true dynamics and predictions of invariant and non-invariant variations of (SWIM) RF-HGN.

## B.2 ROTATION-INVARIANT REPRESENTATION FOR SPATIAL DIMENSION $\mathbf{d = 3}$

The details on obtaining rotation-invariant representations for the spatial dimension $d = 2$ are discussed in Section 3. Here, we extend this approach to the spatial dimension $d = 3$, leveraging the classical Gram-Schmidt orthogonalization method.

We pick $q_1 \in \mathbb{R}^3$ and $q_2 \in \mathbb{R}^3$ as the reference vectors and let $e_1 = \frac{q_1}{\|q_1\|}$ and $e_2' = \frac{q_2}{\|q_2\|}$. $e_1$ is the first basis vector of the new frame. If $|e_1^\mathsf{T} e_2'| > \epsilon$ (near colinear), set $e_2' = e_1 \times e_2'$ (cross product). We then project $u_2 = e_2' - \mathrm{proj}_{e_1} e_2'$ and scale $e_2 = \frac{u_2}{\|u_2\|}$ (Gram-Schmidt) to compute the second basis vector $e_2$, where $\mathrm{proj}_{e_1} e_2' = (e_2'^\mathsf{T} e_1) e_1$ is the projection of $e_2'$ onto $e_1$. We then construct $e_3 = e_1 \times e_2$, where $\times$ is the cross-product. Finally, we define the orthonormal basis $\mathcal{B} = [e_1 \quad e_2 \quad e_3]$. We set $\epsilon = 0.98$ if not specified otherwise.

One can uniquely identify the first two points, independent of node ordering or orientation, as the ones closest to the mean $\bar{q}$. In case of ties, we select the point with the smallest angle relative to the first coordinate axis centered at $\bar{q}$. If the ties persist, we can then select the point with the smallest angle relative to the second coordinate axis.

## B.3 ALGORITHM

Here, we outline the algorithm for the forward pass of the Random-Feature Hamiltonian Graph Network (RF-HGN) using the notation introduced in Section 3.

---

**Algorithm B.1:** Forward pass for RF-HGN: The parameters of all dense layers $\phi_V, \phi_E, \phi_M$ are computed leveraging random sampling techniques and last layer parameters $W_L$ and $b_L$ are computed using least squares (see Section 3.3). We denote the set of neighbors that transmit information to node $j$ by $\mathcal{N}_j$. In the following, we use a single subscript, for instance, for $v_i$, to denote that we compute $v_i$ for all values of $i \in \{1, 2, \ldots, N\}$ for brevity. Also, we use a double subscript, for instance, for $e_{ij}$, to denote that we compute $e_{ij}$ for $i, j \in \{1, \ldots, N\}$ and $i > j$, and set $e_{ji} = e_{ij}$.

---

**Input:** Positions and momenta of the $N$ bodies in spatial dimension $d$ ($p, q \in \mathbb{R}^{2d \cdot N}$), adjacency matrix $A \in \mathbb{R}^{N \times N}$

**Output:** Approximation of Hamiltonian $\widehat{\mathcal{H}} \in \mathbb{R}$

**Parameters:** Node/edge encoder dimension $d_h \in \mathbb{N}$ and message encoder dimension $d_M \in \mathbb{N}$

$\bar{q}_i, \bar{p}_i \in \mathbb{R}^{2d \cdot N} \leftarrow \texttt{encode\_invariances}(p, q)$ for each $i \in \{1, \ldots, N\}$ {Encode translation- and rotation-invariance}

$v_i \leftarrow [\bar{q}_i \quad \bar{p}_i]^\mathsf{T} \in \mathbb{R}^{2 \cdot d}$                                             {Node features}

$e_{ij} \leftarrow \left[(\bar{q}_i - \bar{q}_j)^\mathsf{T}; \|\bar{q}_i - \bar{q}_j\|\right]^\mathsf{T} \in \mathbb{R}^{d+1}$                       {Edge features}

$h_i^V \leftarrow \phi_V(v_i) \in \mathbb{R}^{d_h}$                                                      {Node encoding}

$h_{ij}^E \leftarrow \phi_E(e_{ij}) \in \mathbb{R}^{d_h}$                                                 {Edge encoding}

$h_{ij}^M \leftarrow \phi_M\left(\left[(h_i^V)^\mathsf{T} \quad (h_{ij}^E)^\mathsf{T}\right]^\mathsf{T}\right) \in \mathbb{R}^{d_M}$                    {Message encoding}

$m_j \leftarrow \sum_{i \in \mathcal{N}_j} h_{ij}^M \in \mathbb{R}^{d_M}$                       {Message passing (local pooling)}

$h_G \leftarrow \sum_{j=1}^N \left[(h_j^V)^\mathsf{T} \quad (m_j)^\mathsf{T}\right]^\mathsf{T} \in \mathbb{R}^{d_L}$, where $d_L := d_h + d_M$     {Message passing (global pooling)}

$\widehat{\mathcal{H}} \leftarrow W_L \cdot h_G + b_L$, where $W_L \in \mathbb{R}^{d_L}$ and $b_L \in \mathbb{R}$              {Linear layer}

**return** $\widehat{\mathcal{H}}$

---

The forward pass discussed here is independent of how the network parameters are computed. The training leverages random sampling, automatic differentiation to compute gradients of the Hamiltonian with respect to inputs to compute $\nabla \mathcal{H}$ using PyTorch (Paszke et al., 2019), and least

squares solvers as described in Section 3.3. The network can be formulated more compactly as

$$\widehat{\mathcal{H}}(G) = W_L\left(\sum_{v_j \in V} \phi_V(v_j) \| \sum_{i \in \mathcal{N}_|} \phi_M\Big(\phi_V(v_i), \phi_E(e_{ij})\Big)\right) + b_L, \tag{B.7}$$

where $\|$ is concatenation, given a graph $G = (V, E)$ with a node feature set $V$ and edge feature set $E$ as explained in Section 3 problem setup.

### B.4 Run-time and memory complexity of training

We use the notation defined in Table A.5.

**Run-time complexity:** The bottleneck of the run-time complexity is described in Section 3.3. Encoding the translational symmetry requires a mean-shift of the particles which can be done in $\mathcal{O}(MNd)$ because for each system first the mean value of the positions has to be computed and then the values are updated which can all be done linearly in $M$, $N$, and $d$ given $d$ positions we have to shift. For encoding rotational symmetry we have implemented Gram-Schmidt orthogonalization, which is in $\mathcal{O}(MNd^2)$.

Also note that there are $\frac{M(M-1)}{2}$ pairs of data points to choose from when sampling random features with SWIM. In practice, we do not consider all possible pairs, but rather subsample this set uniformly by choosing the candidate number of pairs to be $\left\lceil \frac{|W|}{M} \right\rceil M$, where $|W|$ is the number of neurons. This is much less than the theoretically possible number of pairs, and still results in a robust sampling method.

**Memory complexity:** Memory requirements for a training set of size $M$ graphs include $\mathcal{O}(d \cdot N \cdot M)$ node features, $\mathcal{O}(d \cdot N_e \cdot M)$ for edge features, and $\mathcal{O}(N_e)$ for the sparse adjacency matrix, assuming the graph stays the same for each example in the training set. For sparsity, we assume $\mathcal{O}(1)$ number of neighbors for each node. The three dense layers (node, edge, and message encoders incur costs of $\mathcal{O}(d_h \cdot d_V)$, $\mathcal{O}(d_h \cdot d_E)$, $\mathcal{O}(d_M \cdot d_h)$. The linear readout layer adds a further $\mathcal{O}(d_L) = \mathcal{O}(d_h + d_M) = \mathcal{O}(d_M)$.

Unlike gradient-descent-based iterative optimization schemes, we only need to compute the gradients of the Hamiltonian $\widehat{\mathcal{H}}$ with respect to inputs, and not with respect to parameters. For this, we additionally need to store the partial derivatives of the output with respect to the input of each dense layer for back-propagation. This amounts to an additional cost of $\mathcal{O}(d_L \cdot d \cdot N \cdot M)$ for the partial derivatives of the global graph value with respect to inputs.

For a fixed spatial dimension $d < 4$ and network width $d_L$, since the dominant terms depend on the dataset size $M$, the number of nodes $N$, and the number of edges in a graph $N_e$, the total memory footprint during training is $\mathcal{O}(M(N + N_e))$. If we further assume zero-shot generalization with a fixed training system, then the total memory requirement is in $\mathcal{O}(M)$ and the geometry of the system (the number of nodes $N$ and edges $N_e$) can grow independently of this training.

## C  Datasets

Table C.6 lists summary information of the datasets used in our experiments, which are explained in more detail in the following subsections. All the constants (masses, spring, and Lennard-Jones constants) are set to one in all the experiments, and for the chain and lattice examples we have used relative distances (all positions are given as displacements relative to the equilibrated state). More information can be found in the code repository: `https://gitlab.com/fd-research/swimhgn`.

Table C.6: Summary of the datasets used in our main experiments. Low and high specify the uniform distribution used to sample the dataset. In Appendix C.1, Appendix C.2 and Appendix C.3 we give more details on how we generate the datasets.

| Experiment | Train points | Test points | Low | High |
|---|---|---|---|---|
| Table 1 | $1000 \cdot N \cdot 6$ | $1000 \cdot N \cdot 6$ | $-0.5$ | $+0.5$ |
| Table 3 | $3000 \cdot N \cdot 4$ | $3000 \cdot N \cdot 4$ | $-1.0$ | $+1.0$ |
| Figure 5 ($N_x$x$N_y$ train) | $1000 \cdot N_x \cdot N_y \cdot 6$ | $1000 \cdot N_x \cdot N_y \cdot 6$ | $-0.5$ | $+0.5$ |
| Figure 5 ($N_x$x$N_y$ test) | — | $1000 \cdot N_x \cdot N_y \cdot 6$ | $-0.5$ | $+0.5$ |
| Figure H.16 and 6 (train $N$) | $2000 \cdot N \cdot 4$ | $2000 \cdot N \cdot 4$ | $-1.0$ | $+1.0$ |
| Figure H.16 (test $N$) | — | $2000 \cdot N \cdot 4$ | $-1.0$ | $+1.0$ |
| Table 2, H.32, H.33; Figure H.18 and H.19 | $810 \cdot N \cdot 4$ | $90 \cdot N \cdot 4$ | $-0.1$ | $+0.1$ |
| Figure H.20 and H.21 | $540 \cdot N \cdot 4$ | $60 \cdot N \cdot 4$ | $-0.1$ | $+0.1$ |
| Section 4.4 | $10000 \cdot N \cdot 4$ | $100 \cdot N \cdot 4$ | — | — |

## C.1 BENCHMARKING AGAINST SOTA OPTIMIZERS

The target Hamiltonian is

$$
\begin{aligned}
\mathcal{H}_1(q,p) = \frac{1}{2}\bigg( & \sum_{i=1}^{Nx}\sum_{j=1}^{Ny}\frac{\|p_{ij}\|^2}{\alpha_{ij}} \\
& + \sum_{i=1}^{Nx}\sum_{j=1}^{Ny-1}\beta_{ij}^x\|q_{i,j+1}-q_{ij}\|^2 + \sum_{j=1}^{Ny}\sum_{i=1}^{Nx-1}\beta_{ij}^y\|q_{i+1,j}-q_{ij}\|^2\bigg),
\end{aligned}
\tag{C.8}
$$

where $q_{ij}, p_{ij} \in \mathbb{R}^3$, and $\alpha_{ij}, \beta_{ij} \in \mathbb{R}$ denote masses and spring constants, respectively. All constants are equal to one if not specified otherwise. $N_x$ and $N_y$ are set to three to build a 3x3 lattice structure (with number of total nodes $N = 9$), which moves in 3D ($d = 3$). We generate a synthetic dataset of 2000 structures (graphs) with their true time derivatives $\{q_i, p_i, \dot{q}_i, \dot{p}_i\}_{i=1}^{2000}$ where $q_i, p_i, \dot{q}_i, \dot{p}_i \in \mathbb{R}^{d \cdot N} \; \forall i$. We first set all $q_i, p_i$ to be in the equilibrium state. Then we sample the displacements $dq_i$ and $dp_i$ from the uniform distribution $U(-0.5, +0.5)$, and compute $q_i \leftarrow q_i + dq_i$ and $p_i \leftarrow p_i + dp_i$. We then compute the ground truths $\dot{q}_i, \dot{p}_i$ using Equation (1) and the ground truth gradient $\nabla\mathcal{H}_1$. We shuffle and split the dataset into train (1000) and test (1000) sets. All the errors reported in Table 1 are the average test errors of three independent runs using different seeds. The total number of training and test points then becomes $1000 \cdot N \cdot d \cdot 2 = 54000$ each.

Additional to the standard spring potential $V(r) = \frac{1}{2}\beta r^2$ given distance $r$ with spring constant $\beta = 1$, we use an anrharmonic spring potential $V(r) = \frac{1}{2}\beta r^2 + \frac{1}{4}\eta r^4$ with nonlinearity coefficient $\eta = 1$ and the Morse potential $V(r) = D(1 - \exp(-ar))^2$ (Morse, 1929) with well-depth $D = 1$ and potential-width $a = 1.0$ for the 2D chain potential experiments in Table 3, (also in Figure H.10, H.11, H.12, H.13, H.14 and H.15). The data generation follows the same procedure explained above, with $N = 5$, resulting in a total of $3000 \cdot N \cdot d \cdot 2 = 60000$ train and test points each.

## C.2 ZERO-SHOT GENERALIZATION AND COMPARISON OF RANDOM FEATURE METHODS

The experiment in Figure 5 uses the same procedure explained in Appendix C.1, with $N_x$ and $N_y$ set to two, three, and four to build 2x2, 3x3, and 4x4 lattice structures.

For the experiment in Figure H.16, the procedure is again similar, but the structure of the experiment and data is different (an open chain). The target function for the open chain system is given in Section 4.2 as

$$
\mathcal{H}_2(q,p) = \frac{1}{2}\bigg(\sum_{i=1}^{N}\frac{\|p_i\|^2}{\alpha_i} + \sum_{i=1}^{N-1}\beta_i\|q_{i+1}-q_i\|^2\bigg),
\tag{C.9}
$$

where $q_i, p_i \in \mathbb{R}^2$, $\alpha_i, \beta_i \in \mathbb{R}$ are positions, momenta, masses, and spring constants in the system, respectively, for $i \in \{1, \ldots, N\}$. All constants are equal to one if not specified otherwise. $\mathcal{H}_2$. $N$ is scaled from exponentially from $2^1$ to $2^{12}$ in the experiment, which always moves in 2D ($d = 2$). For each $N$, we again generate a synthetic dataset of 4000 structures (graphs) with their true time

Table D.7: Model parameters (see Figure 3) used in Section 4.1.

| Model | Encoder width ($d_h$) | Network width ($d_L$) | Activation | Precision |
|---|---|---|---|---|
| HGN (fig. 3) | 48 | 384 | `softplus` | `single` |

Table D.8: Hyperparameters used in Section 4.1 and Section 4.2 are listed for SWIM. `Driver` and `rcond` ($l^2$ reg.) are the parameters of `torch.linalg.lstsq` (Paszke et al., 2019). Resample duplicates specifies to resample till we get a unique pair of points in the SWIM algorithm (Bolager et al., 2023).

| Optimizer | Driver | Parameter sampler | Resample duplicates | $l^2$ reg. |
|---|---|---|---|---|
| SWIM Bolager et al. (2023) | `GELS` | `relu` | `True` | 1e-6 |

derivatives $\{q_i, p_i, \dot{q}_i, \dot{p}_i\}_{i=1}^{4000}$ where $q_i, p_i, \dot{q}_i, \dot{p}_i \; \forall i$. We first set all $q_i, p_i$ to be in the equilibrium state. Then we sample the displacements $dq_i$ and $dp_i$ from the uniform distribution $U(-1.0, +1.0)$, and compute $q_i \leftarrow q_i + dq_i$ and $p_i \leftarrow p_i + dp_i$. We then compute the ground truths $\dot{q}_i, \dot{p}_i$ using Equation (1) and the ground truth gradient $\nabla \mathcal{H}_2$. We shuffle and split the dataset into train (2000) and test (2000) sets.

For the molecular dynamics scenarios with the Lennard-Jones (LJ) potential the Hamiltonian is defined as

$$\mathcal{H}_3(q, p) = \frac{1}{2} \sum_{i=1}^{N} \frac{\|p_i\|}{\alpha_i} + \sum_{i=1}^{N} \sum_{j=i+1}^{N} V^{\text{LJ}}(\|q_j - q_i\|), \qquad \text{(C.10)}$$

where

$$V^{\text{LJ}}(r_{ij}) = 4\epsilon \left[ \left( \frac{\sigma}{r_{ij}} \right)^{12} - \left( \frac{\sigma}{r_{ij}} \right)^{6} \right],$$

and where $r_{ij} = \|q_j - q_i\|$. We set the parameters $\alpha_i, \epsilon, \sigma$ to 1.0 and the cutoff to 2.0 when computing the dynamic edge indices (Figure H.20, H.21), and static edge indices when training with 9 particles and testing with 9 particles (Figure H.18, H.19, Table 2, H.32, H.33). For the static edge experiment, we generated 300 trajectories with 9 particles, and for the dynamic edge experiment, we generated 200 trajectories with 36 particles with the $q$ displacement specified in Table C.6 from the equilibrium state with momenta set to zero. Each trajectory is simulated for 50 time steps with $\Delta t = 5 \cdot 10^{-3}$ and snapshots are taken every $20th$ step with a train-test ratio of 0.9.

### C.3 BENCHMARKING AGAINST SOTA ARCHITECTURES

To benchmark our model, we considered the N-body spring system from Thangamuthu et al. (2022), for which details are available in the original work. Nonetheless, we mention the key properties of the dataset for completeness.

A system of N bodies with equal masses, connected by elastic springs such that each body has two connections and the system forms a closed loop. The system's physical behavior additionally depends on the spring's stiffness and its undeformed length, both are set to one. Initial positions $q_0$ are sampled as $q_0 \sim U(0, 2)$ and initial momenta $p_0$ are sampled as $p_0 \sim U(0, 0.1)$ and subsequently mean-centered. The symplectic Störmer-Verlet (Hairer et al., 2003) integrator with a timestep of $10^{-3}$ is used to generate 100000 datapoints, which are subsampled to 100 datapoints. The approach is repeated for 100 trajectories to obtain a dataset that is split in a 75:25 ratio for a training and validation set. Unlike the original work, the test data we use consists of only one trajectory because with 100 trajectories, we were often experiencing failed simulations with the existing Adam-trained benchmarks (in particular with the LGN architecture), which significantly hinders comparison with our method.

## D TRAINING AND ZERO-SHOT INTEGRATION SETUP

### D.1 BENCHMARKING AGAINST SOTA OPTIMIZERS

Table D.7 and Table D.8 list the model and SWIM hyperparameters, respectively. Table D.9 lists the hyperparameters used for the SOTA optimizers listed in Table 1. All the optimizers are run once

Table D.9: Hyperparameters used in Section 4.1 and Section 4.2 are listed for SOTA optimizers. SGD(+m) represents both SGD Robbins (1951) and SGD+momentum (Sutskever et al., 2013). The `momentum` parameter is set to $0.9$. Avg. SGD specified the Averaged SGD (Gower et al., 2019). Default values are given in the first row (Defaults) for all the optimizers not present in this table, but are listed in Table 1. #steps is the number of total iterations (one iteration per batch). If LR schedule is specified, exponential decay is used as the learning rate scheduler. All optimizers use the `kaiming_normal` (Paszke et al., 2019) weight initialization. $l^2$ regularization ($l^2$ reg.) is specified using the `weight_decay` parameter (Paszke et al., 2019). Full batch size is 1000.

| Optimizer | #steps | Batch size | LR schedule | LR (start, end) | $l^2$ reg. |
|---|---|---|---|---|---|
| Defaults | 10000 | 256 | Yes | 1e-2, 5e-5 | 1e-6 |
| SGD(+m.) | 10000 | 256 | Yes | 5e-4, 5e-5 | 1e-6 |
| Avg. SGD | 10000 | 256 | Yes | 5e-4, 5e-5 | 0 |
| Adadelta | 10000 | 256 | No | 1e-1 (fixed) | 1e-6 |
| Rprop | 2560 | Full | No | 1e-2 (fixed) | 0 |
| RMSprop | 10000 | 256 | Yes | 1e-2, 5e-5 | 0 |
| LBFGS | 100 | Full | No | 1e-1 (fixed) | 0 |

Table D.10: Parameters used in Table 3 (also in Figure H.10, H.11, H.12, H.13, H.14, H.15). #steps is the total number of time steps, $\Delta t$ is the time step size.

| Parameter | Value |
|---|---|
| Symplectic solver | Störmer-Verlet (Hairer et al., 2003) |
| #steps | $1e4$ |
| $\Delta t$ | 1e-2 |

with the default settings that are optimized initially for the Adam optimizer (Kingma & Ba, 2015), and tuned further with multiple iterations. Note that we only want to give a **"time to solution"**, with similar accuracies in order to compare the SOTA optimizers against our method, since the iterative routines can be done arbitrarily long and may be tuned further to reach lower approximation errors than our method with excessive hyperparameter tuning and larger number of iterations for each optimizer–at the cost of even longer training times. **The SGD family** (SGD (Robbins, 1951), SGD+momentum (Sutskever et al., 2013), and Averaged SGD (Gower et al., 2019)) required lower learning rate starts than the adaptive-gradient based optimizers, otherwise they led to `NaN` (Not a Number) results. Even with a very low learning rate, starting at 5e-4, they all produced one `NaN` value out of three experiments, which shows their instability and difficulty in setup. In our results, we therefore only average the two valid results of the SGD-family. **In Averaged SGD** (Gower et al., 2019), the averaging may have acted as an implicit regularizer, and required no weight decay to perform similarly. Also, regularization was not necessary for **Rprop** (Riedmiller & Braun, 1993), **LBFGS** (Liu & Nocedal, 1989), and **RMSprop** (Tieleman & Hinton, 2012). **Adadelta** (Zeiler, 2012) is an adaptive method that dynamically scales updates; therefore, it does not require any scheduler. Also, **Rprop** (Riedmiller & Braun, 1993) uses the sign of the gradients and adapts the step size dynamically, which makes it suitable to be used with large batch updates and no scheduler. Since it uses full batch updates (with batch size of 1000), its number of gradient steps is reduced to provide around the same epoch as the other optimizers. **LBFGS** is a second-order method and outperformed the other optimizers with only 100 steps using full batch updates and no learning rate scheduler.

Table D.11, D.14, D.12, and D.14 list model hyperparameters for the chain potential experiment in Table 3 for training. Table D.10 lists the integration hyperparameters used in the same experiment when zero-shot testing. Note that during testing, we apply a constant gravitational force $[0, -0.075]^{\mathsf{T}}$ to every node in the negative y-axis direction.

## D.2 ZERO-SHOT GENERALIZATION AND COMPARISON OF RANDOM FEATURE METHODS

Table D.15, Table D.8, and Table D.16 list the model, SWIM, and ELM hyperparameters used for the experiments in Section 4.2, respectively. For the zero-shot evaluation presented in Figure 5, we have

Table D.11: Model parameters (see Figure 3) used in Table 3.

| Model | Encoder width ($d_h$) | Network width ($d_L$) | Activation | Precision |
|---|---|---|---|---|
| HGN (fig. 3) | 64 | 1024 | softplus | double |

Table D.12: Hyperparameters used in Table 3 are listed for SWIM. `Driver` and `rcond` ($l^2$ reg.) are the parameters of `torch.linalg.lstsq` (Paszke et al., 2019). Resample duplicates specifies to resample till we get a unique pair of points in the SWIM algorithm (Bolager et al., 2023).

| Optimizer | Driver | Parameter sampler | Resample duplicates | $l^2$ reg. |
|---|---|---|---|---|
| SWIM Bolager et al. (2023) | GELS | relu | True | 1e-10 |

Table D.13: Hyperparameters used in Table 3 are listed for ELM (Huang et al., 2004). `Driver` and `rcond` ($l^2$ reg.) are the parameters of `torch.linalg.lstsq` (Paszke et al., 2019). Bias low and high specify the uniform distribution of low and high values, from which the biases of the random feature layers are sampled. The weights are sampled using the standard normal distribution as explained in Section 3.3.

| Optimizer | Driver | Bias low | Bias high | $l^2$ reg. |
|---|---|---|---|---|
| ELM (Huang et al., 2004) | GELS | -1.0 | +1.0 | 1e-10 |

trained (SWIM) RF-HGN ten times with different random seeds (also see Figure H.8, Figure H.9, and Table H.31), and used the pretrained (SWIM) RF-HGN model with the median test error to evaluate on the zero-shot test cases in order to avoid any statistical bias, as this is a random feature method. Table D.17 lists the parameters used to integrate the system in Figure 6 and Figure H.17.

For the molecular systems, Table D.19 lists the model parameters, Table D.20 lists SWIM hyperparameters, Table D.21 lists ELM hyperparameters, and Table D.22 lists Adam hyperparameters. No early stopping was triggered in these experiments. Table D.18 lists the parameters used to integrate all the molecular systems presented in this paper.

### D.3 BENCHMARKING AGAINST SOTA ARCHITECTURES

Table D.23 lists the model parameters, Table D.24 lists SWIM hyperparameters, Table D.25 lists SOTA architecture hyperparameters used in Section 4.4.

## E HARDWARE

Table E.26 lists hardware used for all the experiments presented in Section 4. The experiments presented in section 4.1, Figure 5 (training of the 2x2 and 3x3 lattice systems), and Section 4.4 are conducted on a CUDA GPU. Figure 5 (training of the 4x4 lattice), Figure 5 (testing), and Figure H.16 are conducted on CPUs because of their larger memory requirements.

## F ABLATION STUDIES

We vary the widths of the encoders and linear layers to understand how they affect the mean squared error defined on the true and predicted solutions. We experimented with the Hamiltonian $\mathcal{H}_2$ explained in Section 4.2, chain of 8 nodes in 2D. We evaluated the model on the training and testing sets with 2000 samples each in the phase space and report the test errors. The message encoder's width is chosen by subtracting the width of the hidden layer from the width of the linear layer in the ablation study, i.e., $d_M = d_L - d_h$. Figure F.7 and Table F.27 reveal that increasing either the linear layer width $d_L$ or the hidden dimension $d_h$ while keeping the other parameter fixed consistently reduces the mean squared error. In the same experiment, we also computed the condition number $\kappa(Z) = \frac{\sigma_1}{\sigma_n}$ in terms of the singular values $\sigma_1 > \cdots > \sigma_n$ of the matrix $Z$ associated with the linear system in Equation (6) to assess the sensitivity of the solution. We avoided the bias term when computing the condition, as it was only used to fit the integration constant in practice. Table F.28 reveals larger values

Table D.14: Hyperparameters used in Table 3 are listed for Adam.

| Optimizer | #steps | Batch size | LR schedule | LR (start, end) | $l^2$ reg. | Initialization | Patience |
|---|---|---|---|---|---|---|---|
| Adam | 10000 | 256 | exponential decay | 1e-2, 5e-5 | 1e-6 | Kaiming normal | 1000 |

Table D.15: Model parameters used in Section 4.2. The network width specifies the size of the input to the last linear layer in both RF-HNN (Bertalan et al., 2019; Greydanus et al., 2019) and RF-HGN.

| Model | Encoder width ($d_h$) | Network width | Activation | Precision |
|---|---|---|---|---|
| RF-HGN (fig. 3) | 64 | 512 | softplus | single |
| RF-HNN | — | 512 | softplus | single |

Table D.16: Hyperparameters used in Section 4.2 are listed for ELM (Huang et al., 2004). `Driver` and `rcond` ($l^2$ reg.) are the parameters of `torch.linalg.lstsq` (Paszke et al., 2019). Bias low and high specify the uniform distribution of low and high values, from which the biases of the random feature layers are sampled. The weights are sampled using the standard normal distribution as explained in Section 3.3.

| Optimizer | Driver | Bias low | Bias high | $l^2$ reg. |
|---|---|---|---|---|
| ELM (Huang et al., 2004) | GELS | -1.0 | +1.0 | 1e-6 |

Table D.17: Parameters used in Figure 6 (and consequently Figure H.17). #steps is the total number of time steps, $\Delta t$ is the time step size.

| Parameter | Value |
|---|---|
| Symplectic solver | Störmer-Verlet (Hairer et al., 2003) |
| #steps | 5000 |
| $\Delta t$ | 1e-3 |

Table D.18: Parameters used in Figure H.18 and Figure H.20 (and consequently Table 2, H.32, H.33, Figure H.19 H.21). #steps is the total number of time steps, $\Delta t$ is the time step size.

| Parameter | Value |
|---|---|
| Symplectic solver | Störmer-Verlet (Hairer et al., 2003) |
| #steps | $1e5$ |
| $\Delta t$ | 1e-5 |

Table D.19: Model parameters used in Table 2, H.32, H.33, Figure H.18, H.19 (top row) and in Figure H.20, H.21 (bottom row) are listed for the RF-HGNs and Adam-HGN.

| Model | Encoder width ($d_h$) | Network width ($d_L$) | Activation | Precision |
|---|---|---|---|---|
| HGN | 40 | 800 | softplus | single |
| HGN | 32 | 256 | gelu | single |

Table D.20: Hyperparameters used in Table 2, H.32, H.33, Figure H.18, H.19 and in Figure H.20, H.21 are listed for SWIM. `Driver` and `rcond` ($l^2$ reg.) are the parameters of `torch.linalg.lstsq` (Paszke et al., 2019). Resample duplicates specifies to resample till we get a unique pair of points in the SWIM algorithm (Bolager et al., 2023).

| Optimizer | Driver | Parameter sampler | Resample duplicates | $l^2$ reg. |
|---|---|---|---|---|
| SWIM Bolager et al. (2023) | GELSD | relu | True | 1e-10 |

Table D.21: Hyperparameters used in Table 2, H.32, H.33, Figure H.18, H.19 and in Figure H.20, H.21 are listed for ELM (Huang et al., 2004). `Driver` and `rcond` ($l^2$ reg.) are the parameters of `torch.linalg.lstsq` (Paszke et al., 2019). Bias low and high specify the uniform distribution of low and high values, from which the biases of the random feature layers are sampled. The weights are sampled using the standard normal distribution as explained in Section 3.3.

| Optimizer | Driver | Bias low | Bias high | $l^2$ reg. |
|---|---|---|---|---|
| ELM (Huang et al., 2004) | GELSD | -1.0 | +1.0 | 1e-10 |

Table D.22: Hyperparameters used in Table 2, H.32, H.33, Figure H.18, H.19 and in Figure H.20, H.21 are listed for Adam.

| Optimizer | #steps | Batch size | LR schedule | LR (start, end) | $l^2$ reg. | Initialization | Patience |
|---|---|---|---|---|---|---|---|
| Adam | 10000 | 8 | exponential decay | 1e-3, 5e-5 | 1e-10 | Kaiming normal | 500 |

Table D.23: Random feature Hamiltonian graph network (RF-HGN) parameters used in Section 4.4. The network width specifies the size of the input to the last linear layer in the RF-HGN.

| Model | Encoder width ($d_h$) | Network width | Activation | Precision |
|---|---|---|---|---|
| RF-HGN (fig. 3) | 32 | 512 | softplus | double |

Table D.24: Hyperparameters used in Section 4.4 are listed for SWIM (Bolager et al., 2023). `Driver` and `rcond` ($l^2$ reg.) are the parameters of `torch.linalg.lstsq` (Paszke et al., 2019).

| Optimizer | Driver | Parameter sampler | Resample duplicates | $l^2$ reg. |
|---|---|---|---|---|
| SWIM (Bolager et al., 2023) | GELS | relu | True | 1e-15 |

Table D.25: Hyperparameters used in Section 4.4 are listed.

| Model | FGNN, FGNODE, LGN, HGN | GNODE, LGNN, HGNN |
|---|---|---|
| Node embedding dim. | 8 | 5 |
| Edge embedding dim. | 8 | 5 |
| # hidden layers | 2 | 2 |
| # hidden neurons (per layer) | 16 | 5 |
| # message passing layers | 1 | 1 |
| Activation | squareplus | squareplus |
| Optimizer | Adam | Adam |
| Learning rate | $10^{-3}$ | $10^{-3}$ |
| Batch size | 100 | 100 |
| Epochs | 10000 | 10000 |
| Precision | double | double |
| $l^2$ regularization | — | — |

Table E.26: Hardware used for the experiments is listed with details on CPUs (Intel i7 and AMD EPYC), memory, GPU (NVIDIA), CUDA version (driver version, CUDA version), and operating system (OS) versions of Ubuntu LTS, together with memory requirements (in GB).

| Experiment | CPU (cores) | Memory | GPU (vram) | CUDA | OS |
|---|---|---|---|---|---|
| section 4.1 | i7-14700K (20) | 66 | RTX 4070 (12) | 550.120, 12.4 | 24.04.2 |
| fig. 5 train | i7-14700K (20) | 66 | RTX 4070 (12) | 550.120, 12.4 | 24.04.2 |
| fig. 5 4x4 train | i7-14700K (20) | 66 | — | — | 24.04.2 |
| fig. 5 test | EPYC 7402 (24) | 256 | — | — | 20.04.2 |
| fig. H.16 | EPYC 7402 (24) | 256 | — | — | 20.04.2 |
| section 4.2 molecular systems | i7-14700K (20) | 66 | — | — | 24.04.2 |
| section 4.4 | EPYC 7402 (24) | 256 | — | — | 20.04.2 |

Table F.27: Ablation study showing the influence of widths of the dense layers encoding the node and edge features and the linear layer on the mean squared error between predicted and true system dynamics $\dot{y}$.

| | $d_L = 128$ | $d_L = 256$ | $d_L = 512$ |
|---|---|---|---|
| $d_h = 8$ | 2.879e-02 | 1.336e-02 | 3.666e-03 |
| $d_h = 16$ | 1.657e-03 | 1.634e-04 | 6.571e-05 |
| $d_h = 32$ | 8.290e-04 | 2.011e-04 | 1.278e-05 |
| $d_h = 64$ | 7.826e-04 | 3.037e-05 | 7.713e-06 |

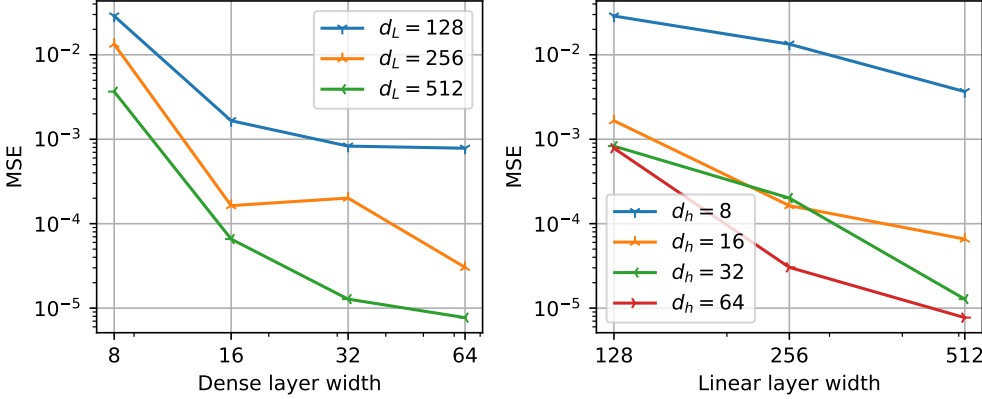

Figure F.7: Ablation study for widths of the dense and linear layers.

Table F.28: Ablation study showing the condition number of the matrix associated with Equation (6) with increasing feature widths.

|  | $d_L = 128$ | $d_L = 256$ | $d_L = 512$ |
|---|---|---|---|
| $d_h = 8$ | 4.266e+05 | 8.985e+06 | 2.612e+08 |
| $d_h = 16$ | 1.525e+05 | 1.687e+06 | 3.545e+07 |
| $d_h = 32$ | 8.105e+04 | 1.352e+06 | 2.974e+07 |
| $d_h = 64$ | 1.072e+05 | 9.625e+05 | 1.766e+07 |

Table F.29: Ablation study showing the influence of applying recursive message-passing on the mean squared error between predicted and true system dynamics $\dot{y}$. #msg is the number of message passes.

|  | #msg $= 1$ | #msg $= 2$ | #msg $= 3$ | #msg $= 4$ | #msg $= 5$ | #msg $= 6$ |
|---|---|---|---|---|---|---|
| Summing | 7.713e-06 | 1.866e-05 | 1.271e-04 | 1.150e-03 | 7.620e-04 | 2.222e-03 |
| Averaging | 2.020e-02 | 1.265e-02 | 1.280e-02 | 1.057e-02 | 1.175e-02 | 1.089e-02 |

as the system size (network width $d_L$) increases, but increasing the feature width of the encoders $d_h$ slightly stabilizes the system for large network widths, as expected.

We additionally vary the number of message passes (#msg) by recursively applying the local-pooling $h_j^V \leftarrow \sum_{i \in \mathcal{N}_j} h_i^V$ aggregating the node encodings $h_i^V$ of the neighboring nodes $i \in \mathcal{N}_j$ (#msg$-1$ times), and then applying the final message scheme explained in Section 3.2.2 with two different schemes: summing ($h_j^V \leftarrow \sum_{i \in \mathcal{N}_j} h_i^V$) and averaging ($h_j^V \leftarrow \frac{1}{|\mathcal{N}_j|} \sum_{i \in \mathcal{N}_j} h_i^V$). Table F.29 reveals that having multiple message-passes can improve the accuracy for the 8-particle mass-spring system when averaging is used. We believe that summing works better than averaging because it implicitly encodes the node degree information by aggregating the neighboring messages. Each neighboring message is the output of a `softplus` activation function and has non-negative values. In all the other experiments presented in this paper, we use only a single message pass and do not optimize the number of message passes, as all the ground truth systems we consider only require a single step of neighborhood information.

# G  COMPARISON WITH A BENCHMARK DATASET

To further support our claims, here we perform benchmarking of our model against existing suitable graph network approaches. We made use of the existing publication from the NeurIPS 2022 Datasets and Benchmarks Track by Thangamuthu et al. (2022) and their corresponding repository. The considered models for comparison include:

- **Full Graph Neural Network (FGNN)** : Based on the work of Sanchez-Gonzalez et al. (2020), these models utilize message-passing as a key feature to enable a simulation framework. Note that in the original work the architecture is called Graph Network-based Simulators (GNS) but for benchmarking it is called FGNN and we use this name as well.

- **Full Graph Neural ODE (FGNODE)** : An ODE version of FGNN is what we refer to as FGNODE (Sanchez-Gonzalez et al., 2019).

- **Graph Neural ODE (GNODE)** : This architecture uses a graph topology to parameterize the force of a system using a neural ODE approach, it was introduced by Thangamuthu et al. (2022).

- **Lagrangian Graph Network (LGN)** : This architecture uses an FGNN to predict the Lagrangian of the system (Bhattoo et al., 2022).

- **Lagrangian Graph Neural Network (LGNN)** : Similar to LGN, this architecture decoples the kinetic and potential energies (Thangamuthu et al., 2022).

- **Hamiltonian Graph Network (HGN)** : In this architecture an FGNN predicts the Hamiltonian of the system (Sanchez-Gonzalez et al., 2019; Thangamuthu et al., 2022).

- **Hamiltonian Graph Neural Netwrok (HGNN)** : Analogously, this architecture is similar to HGN but it decouples the potential and kinetic energies of the Hamiltonian (Thangamuthu et al., 2022).

Table G.30: Comparison of the SOTA physics-informed graph network architectures (also see Table 4) and our (SWIM) RF-HGN.

| Model | (SWIM) RF-HGN | FGNN | FGNODE | GNODE | LGN | LGNN | HGN | HGNN |
|---|---|---|---|---|---|---|---|---|
| Translation invariance | ✓ | ✓ | ✗ | ✗ | ✗ | ✗ | ✗ | ✗ |
| Rotation invariance | ✓ | ✗ | ✗ | ✗ | ✗ | ✗ | ✗ | ✗ |
| Energy conservation | ✓ | ✗ | ✗ | ✗ | ✓ | ✓ | ✓ | ✓ |
| Gradient-descent-free training | ✓ | ✗ | ✗ | ✗ | ✗ | ✗ | ✗ | ✗ |

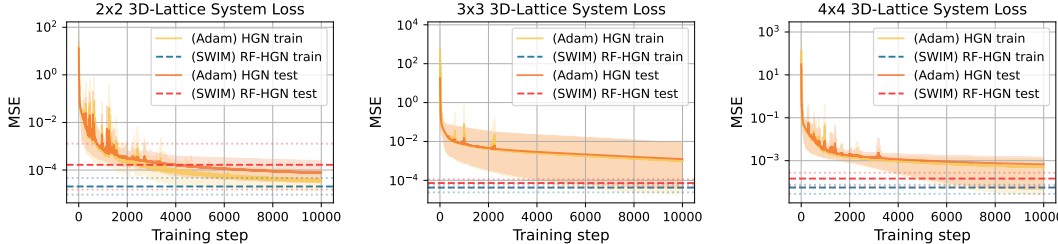

Figure H.8: MSE losses on the training and test dataset for a 2x2 (left), 3x3 (middle) and 4x4 (right) lattice during iterative training are given with solid lines for the average over ten runs; the shaded region extends from the minimum to the maximum value. The dashed lines denote the (constant) MSE losses for our non-iterative optimization, and shaded dashes show the minimum and maximum.

First, we highlight the similarities and differences in model properties in table G.30, noting that our model satisfies requirements necessary for modeling physical systems while maintaining energy conservation.

A key metric of interest for our work is the training time, thus we have re-trained models from (Thangamuthu et al., 2022) on their datasets of the spring system with 3, 4 and 5 nodes and record the training time. All runs were performed on the same machine as the experiments in the main paper. The resulting training times in table Table 4 show clearly that our proposed approach is much faster to train, especially compared to the specialized Lagrangian and Hamiltonian graph networks.

Of course, a model is only useful if it can accurately make predictions, thus we plot errors on a test trajectory for all mentioned models in Figure H.22. We observe that our (SWIM) RF-HGN has a similar predictive ability as the SOTA architectures. It should be noted that for the test trajectory shown for $N = 4$ the LGN model diverged after around 25 steps. Similar results of diverging models were also observed from LGNN and the NODE architectures when we attempted to test on 100 trajectories, where multiple predicted trajectories would diverge from the true trajectory.

# H    ADDITIONAL RESULTS

Our submitted folder contains an animation of the test system shown in Figure 1 of the main text, as well as the molecular dynamics systems (see Figure H.18 and Figure H.20).

## H.1    BENCHMARKING AGAINST SOTA OPTIMIZERS

In Figure H.8, we show loss curves for the Adam optimizer, highlighting how its train and test losses evolve over time relative to the loss of our non-iterative approach in 2x2, 3x3, and 4x4 systems. The model and optimizer hyperparameters are set accordingly as explained in Appendix C.1, Table D.7, Table D.9, and Table D.8. We observe comparable accuracies of Adam and SWIM (Bolager et al., 2023), even after 10000 gradient descent iterations using the Adam (Kingma & Ba, 2015) optimizer. Moreover, Figure H.9 reveals that our method scales better than iterative optimization, maintaining low error as system size increases. And Table H.31 reveals two to three orders of magnitude quicker training of (SWIM) RF-HGN than (Adam) HGN in different 3D lattice systems.

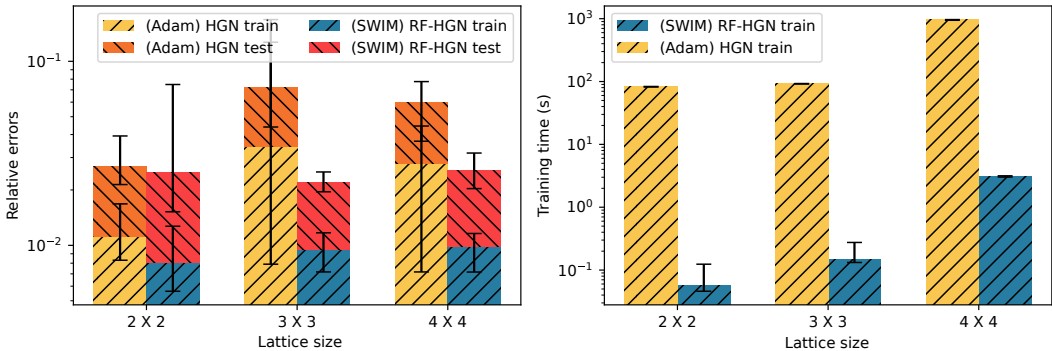

Figure H.9: Relative error and training time are shown for different lattice sizes. Boxplots show the mean and error bars based on ten runs with different random seeds.

Table H.31: Summary of the training times of the experiment presented in Figure H.9 for (SWIM) RF-HGN and (Adam) HGN in seconds. For the systems of sizes $2 \times 2$ (GPU trained), $3 \times 3$ (GPU trained), and $4 \times 4$ (CPU trained), we observe approximately three, two, and three orders of magnitude faster training, respectively.

| System size | (SWIM) RF-HGN | (Adam) HGN |
|---|---|---|
| $2 \times 2$ | $\approx 0.06$ seconds | $\approx 82.93$ seconds |
| $3 \times 3$ | $\approx 0.15$ seconds | $\approx 92.1$ seconds |
| $4 \times 4$ | $\approx 3.06$ seconds | $\approx 936.12$ seconds |

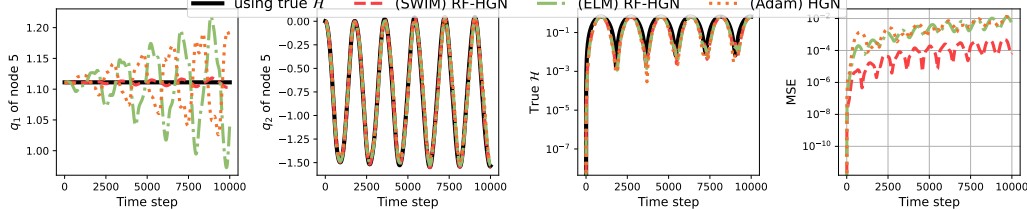

Figure H.10: Illustration of position trajectories of the middle node over time (also see Figure H.11). Models are trained with 5 nodes and tested (here) with 10 nodes using the standard spring potential (see Appendix C.1).

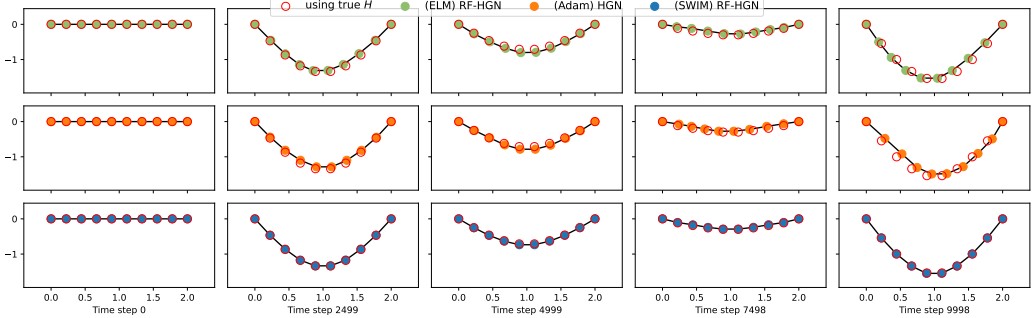

Figure H.11: Illustration of position trajectories over time from models trained on a system with 5 nodes using the spring chain potential (see Appendix C.1) and zero-shot tested with 10 nodes with an external force (gravitational).

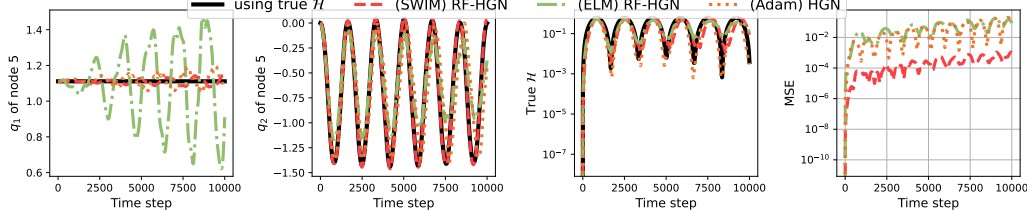

Figure H.12: Illustration of position trajectories of the middle node over time (also see Figure H.11). Models are trained with 5 nodes and tested (here) with 10 nodes using anharmonic spring potential (see Appendix C.1).

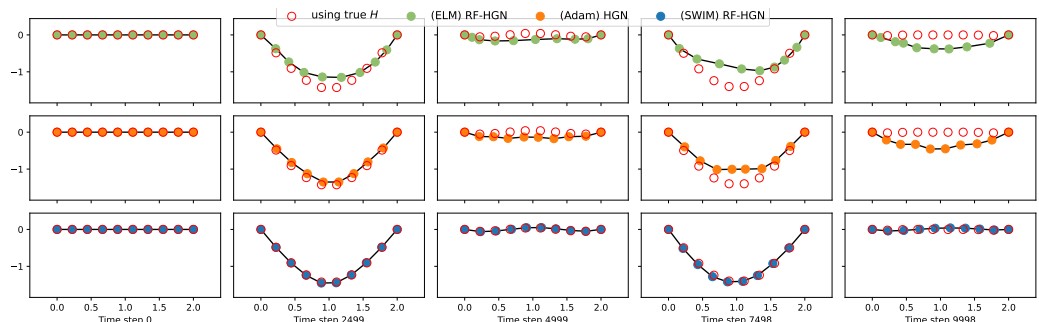

Figure H.13: Illustration of position trajectories over time from models trained on a system with 5 nodes using anharmonic spring potential (see Appendix C.1) and zero-shot tested with 10 nodes with an external force (gravitational).

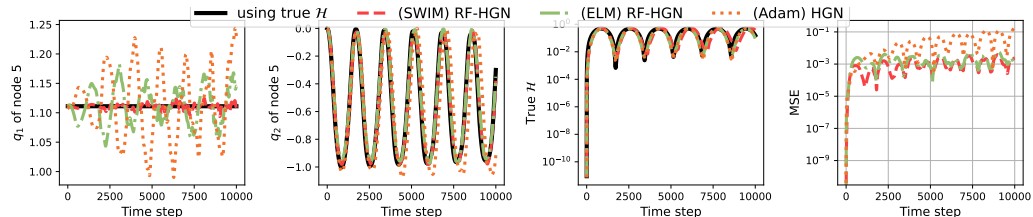

Figure H.14: Illustration of position trajectories of the middle node over time (also see Figure H.11). Models are trained with 5 nodes and tested (here) with 10 nodes using the Morse potential (see Appendix C.1).

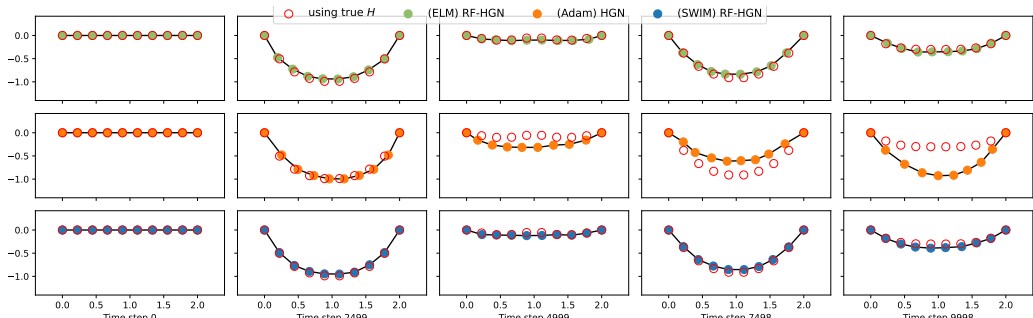

Figure H.15: Illustration of position trajectories over time from models trained on a system with 5 nodes using the Morse spring potential (see Appendix C.1) and zero-shot tested with 10 nodes with an external force (gravitational).

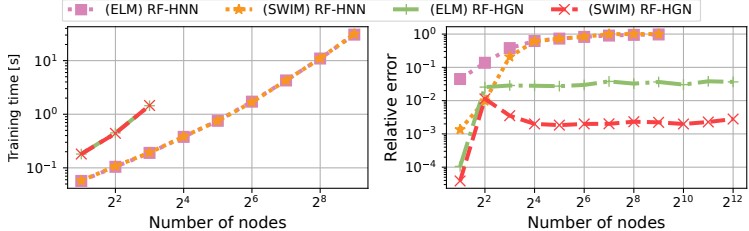

Figure H.16: Zero-shot generalization in 2D open chain (see Figure 4 (b)): RF-HGN trained up to $N = 8$ accurately generalizes up to $N = 4096$, outperforming retrained RF-HNN (right). RF-HGN with zero-shot generalization is also faster than RF-HNN for node counts larger than $2^6$ (left).

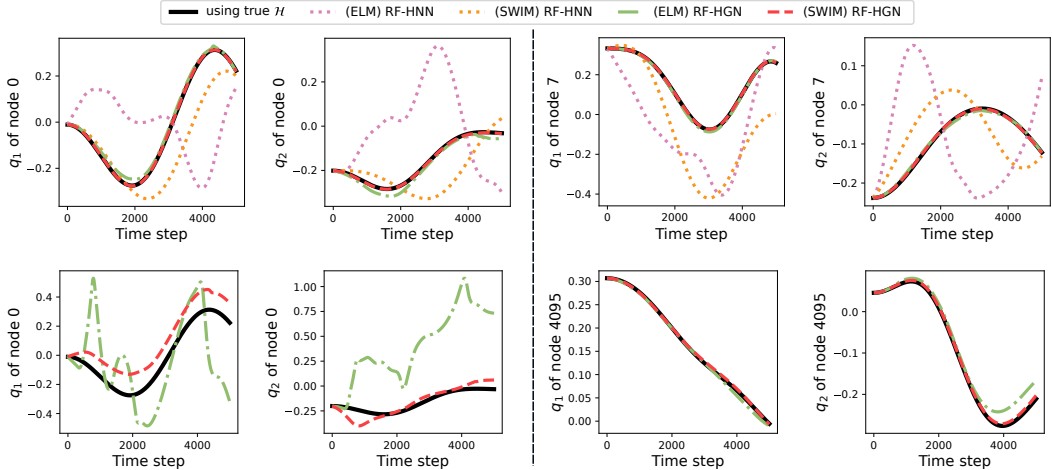

Figure H.17: Illustration of position trajectories of the corner nodes over time (also see Figure 6). Top: Trained with $2^3$ nodes and tested with $2^3$ nodes. Bottom: Trained with $2^3$ nodes and tested with $2^{12}$ nodes.

## H.2 ZERO-SHOT GENERALIZATION AND COMPARISON OF RANDOM FEATURE METHODS

Figure H.17 illustrates the trajectories of the corner nodes of the experiment in Figure 6. For this particular example the left corner trajectory (node with id 0) seems to be harder to capture than the other nodes in the system for the extreme zero-shot case (trained with 8 nodes, tested with 4096 nodes) case, hence slightly higher error on the trajectories compared to the non-zero-shot case (trained with 8 nodes and tested with 8 nodes) as one can see in Figure 6.

## H.3 BENCHMARKING AGAINST SOTA ARCHITECTURES

## H.4 ROBUSTNESS AGAINST NOISE

To evaluate our model's robustness against additive noise, we add Gaussian noise with different standard deviations ($\sigma$) to the ground truth positions and momenta before including them in the training set. We experimented with 5 nodes in 3D on the training set with 1000 samples in phase

Table H.32: Molecular dynamics evaluation with 9 particles. Mean squared error (MSE) and relative $l^2$ error (rel. $l^2$) are reported together with the true Hamiltonian over the ground-truth trajectory and the (ELM) RF-HGN predicted quantity over the rolled-out trajectory.

|  | T=1 | T=25000 | T=50000 | T=74999 | T=99999 |
|---|---|---|---|---|---|
| $q$ MSE | 8.651e-08 | 9.369e+07 | 2.926e+09 | 9.456e+08 | 6.337e+09 |
| $q$ rel. $l^2$ | 2.044e-04 | 6.766e+03 | 3.718e+04 | 2.139e+04 | 5.615e+04 |
| True $\mathcal{H}$ | -1.233e+01 | -1.233e+01 | -1.233e+01 | -1.233e+01 | -1.233e+01 |
| Model $\widehat{\mathcal{H}}$ | -1.450e+01 | 2.979e+10 | -1.122e+10 | -3.085e+10 | 8.765e+10 |

Table H.33: Molecular dynamics evaluation with 9 particles. Mean squared error (MSE) and relative $l^2$ error (rel. $l^2$) are reported together with the true Hamiltonian over the ground-truth trajectory and the (Adam) HGN predicted quantity over the rolled-out trajectory.

|  | T=1 | T=25000 | T=50000 | T=74999 | T=99999 |
|---|---|---|---|---|---|
| $q$ MSE | 3.089e-13 | 5.043e-03 | 1.554e-02 | 6.052e-02 | 1.279e-01 |
| $q$ rel. $l^2$ | 3.863e-07 | 4.964e-02 | 8.568e-02 | 1.711e-01 | 2.522e-01 |
| True $\mathcal{H}$ | -1.233e+01 | -1.233e+01 | -1.233e+01 | -1.233e+01 | -1.233e+01 |
| Model $\widehat{\mathcal{H}}$ | -1.233e+01 | -1.233e+01 | -1.233e+01 | -1.233e+01 | -1.233e+01 |

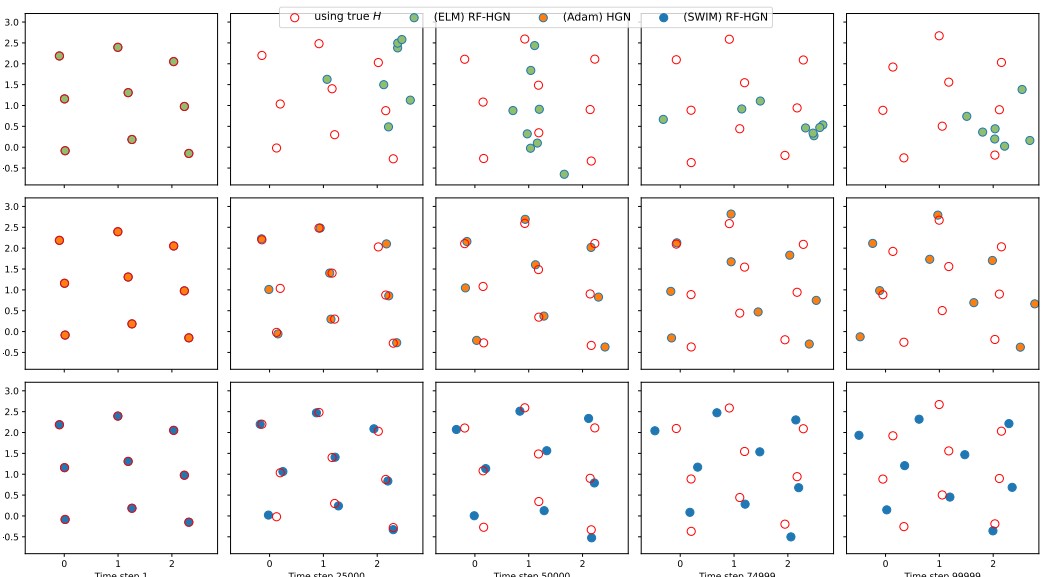

Figure H.18: Illustration of position trajectories over time from models trained on a system with 9 nodes on the 2D molecular dynamics system (see Figure 4 (c)).

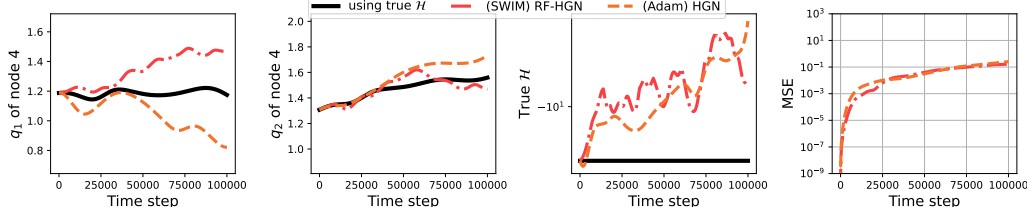

Figure H.19: Illustration of position trajectories over time from models trained on a system with 9 nodes on the 2D molecular dynamics system (see Figure 4 (c)). Results from ELM RF-HGN training are omitted due to very large errors which distort the representations in the plots.

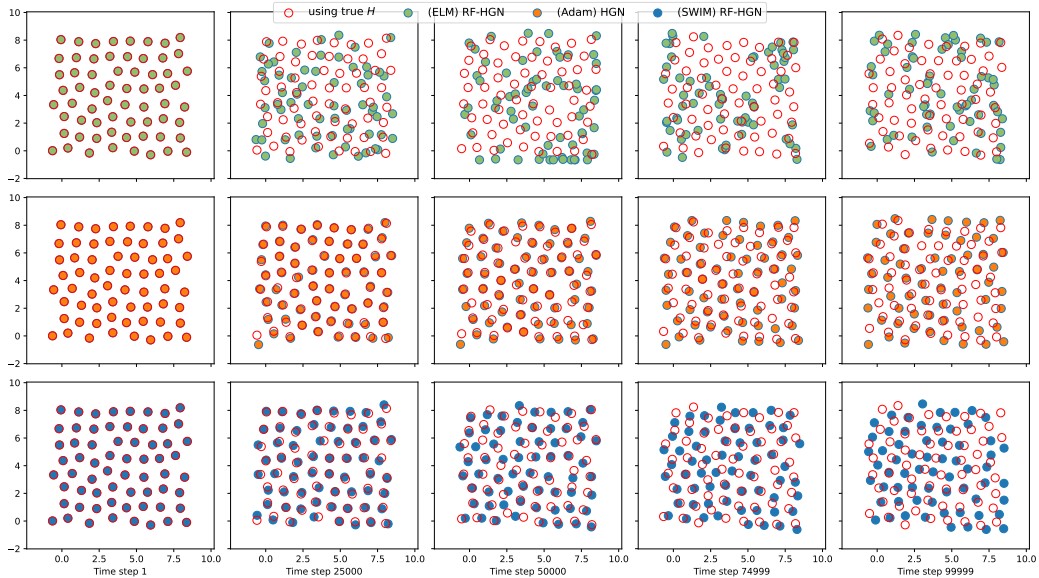

Figure H.20: Illustration of position trajectories over time from models trained on a system with 36 nodes and zero-shot tested with 64 nodes on the 2D molecular dynamics system (see Figure 4 (d)).

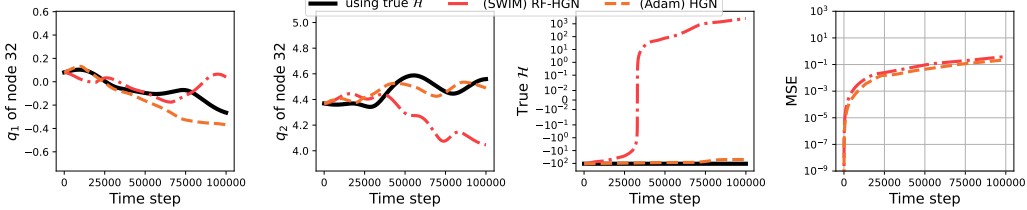

Figure H.21: Illustration of position trajectories over time from models trained on a system with 36 nodes and tested with 64 nodes on the 2D molecular dynamics system (see Figure 4 (c)). Results from ELM RF-HGN training are omitted due to very large errors which distort the representation in the plots.

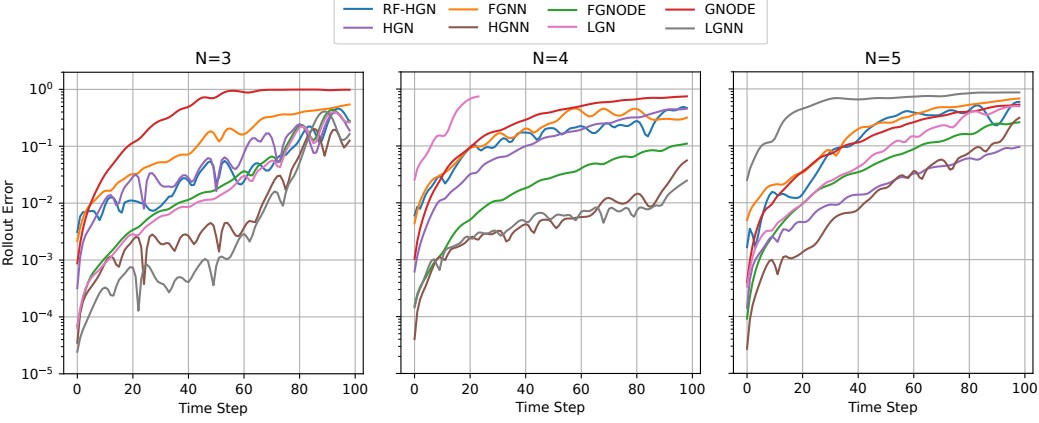

Figure H.22: Rollout errors on test trajectory benchmark for $N = 3, 4, 5$.

Table H.34: Results of training with noisy data are displayed. Relative $l^2$ and mean-squared errors are displayed.

| $\sigma$ | 0 | 1e-5 | 1e-4 | 1e-3 | 1e-2 | 1e-1 | 1 | 2 |
|---|---|---|---|---|---|---|---|---|
| CV MSE | 6.18e-3 | 6.59e-3 | 5.35e-3 | 7.84e-3 | 6.45e-3 | 3.38e-2 | 2.78 | 1.11e+1 |
| CV rel. $l^2$ | 4.03e-2 | 4.07e-2 | 3.72e-2 | 4.29e-2 | 4.05e-2 | 9.61e-2 | 6.64e-1 | 8.78e-1 |
| Test MSE | 9.52e-3 | 9.77e-3 | 9.12e-3 | 1.04e-2 | 9.12e-3 | 9.67e-3 | 1.01e-1 | 3.94e-1 |
| Test rel. $l^2$ | 4.96e-2 | 4.92e-2 | 4.83e-2 | 5.09e-2 | 4.80e-2 | 4.94e-2 | 1.65e-1 | 3.27e-1 |

Table H.35: Batch-wise training results compared to direct least squares solutions are displayed with training time in seconds and memory usage in GiB. Relative $l^2$ and mean-squared errors are displayed.

| | ELM | ELM (batched) | SWIM | SWIM (batched) |
|---|---|---|---|---|
| CV MSE | 9.01 | 2.15e+03 | 3.84e-1 | 1.08 |
| CV rel. $l^2$ | 2.59e-1 | 2.54 | 4.65e-2 | 7.89e-2 |
| Test MSE | 9.06 | 1.98e+3 | 3.71e-1 | 1.09 |
| Test rel. $l^2$ | 2.6e-1 | 2.49 | 4.61e-2 | 8.02e-2 |
| Training time | 3.96 | 4.48 | 3.88 | 4.42 |
| Memory usage | 2.4 | 1.1 | 2.4 | 1.1 |

space using 5-fold cross-validation, and on a test set of 1000 samples in phase space. To stabilize the results, we repeated this experiment 5 times with different seeds and report the average results in Table H.34.

The columns in Table H.34 list the standard deviation $\sigma$ used in the experiments. The results indicate the robustness of our method to Gaussian noise added to the state. For more realistic scenarios, the model can further be improved by modeling uncertainties and assessing sensitivities for noise in a real-data setting.

## H.5 BATCH-WISE TRAINING

To demonstrate batching, we prepared an example with Lennard-Jones potential with 4 particles in 2D trained with both ELM, SWIM, and their batch-wise versions when solving the linear problem (Equation (6)). Batching is performed by sub-sampling the training data set (5000 states in phase space) and averaging the resulting last-layer coefficients. Table H.35 lists the results, where the columns indicate which random feature method is used, rows indicate cross-validation (5-fold) errors with train set size 5000 and test errors averaged over 5 different seeds of test size 5000.

We note that the memory required for batch-wise training can be further improved with additional tuning, for example, by explicitly freeing the GPU memory, we could tune the memory requirement to be as minimal as possible (0.04 GiB). In this case, however, the training time also increases to around $5.7$ seconds. Different tuning strategies (e.g., for lower memory, for quicker runtime) are therefore important to consider when comparing different training strategies.

We believe that with an established linear solver like LSQR (Paige & Saunders, 1982), LSMR (Fong & Saunders, 2011), LSRN (Meng et al., 2014a), one can further study the batch-wise training of our model for even larger systems, tuned for specific needs such as low memory or fast training.

## H.6 BENCHMARKING DIFFERENT RANDOM FEATURES

We also experimented with random Fourier features (RFF) (Rahimi & Recht, 2007) by setting

$$W_{ij} \sim \mathcal{N}(0, \frac{1}{\sigma^{\mathrm{RFF}}}), \quad b_i \sim \mathtt{Uniform}(0, 2\pi), \quad z = \sqrt{\frac{2}{\mathtt{\#features}}} \cos(W^\mathsf{T} x + b),$$

where $z$ is the random features and $\mathtt{\#features}$ is the size of $z$.

We have extended the noise-scale experiment in Appendix H.4 with $\sigma^{\mathrm{RFF}} = 1$ and list the results in table Table H.36. Additionally, we run the same experiment using ELM by setting

$$W_{ij} \sim \mathcal{N}(0, 1), \quad b_i \sim \mathtt{Uniform}(-\pi, \pi),$$

and list the results in Table H.37

Table H.36: Results of training with noisy data using RFF are displayed. Relative $l^2$ and mean-squared errors are displayed.

| $\sigma$ | 0 | 1e-5 | 1e-4 | 1e-3 | 1e-2 | 1e-1 | 1 | 2 |
|---|---|---|---|---|---|---|---|---|
| CV MSE | 3.02 | 3.03 | 3.05 | 3.04 | 3.01 | 3.09 | 5.59 | 1.4e+1 |
| CV rel. $l^2$ | 9.1e-1 | 9.14e-1 | 9.15e-1 | 9.13e-1 | 9.1e-1 | 9.17e-1 | 9.41e-1 | 9.87e-1 |
| Test MSE | 3.43 | 3.61 | 3.44 | 3.47 | 3.48 | 3.39 | 3.62 | 3.87 |
| Test rel. $l^2$ | 9.74e-1 | 1 | 9.77e-1 | 9.82e-1 | 9.84e-1 | 9.71e-1 | 1 | 1.04 |

Table H.37: Results of training with noisy data using ELM are displayed. Relative $l^2$ and mean-squared errors are displayed.

| $\sigma$ | 0 | 1e-5 | 1e-4 | 1e-3 | 1e-2 | 1e-1 | 1 | 2 |
|---|---|---|---|---|---|---|---|---|
| CV MSE | 3e-1 | 3.2 | 2.44e-1 | 6.78e-1 | 2.4e-1 | 4.62e-1 | 8.75 | 2.32e+2 |
| CV rel. $l^2$ | 2.54e-1 | 5.22e-1 | 2.39e-1 | 3.24e-1 | 2.31e-1 | 2.87e-1 | 8.73e-1 | 1.82 |
| Test MSE | 4.7e-1 | 2.89e-1 | 2.63e-1 | 3.65e-1 | 3.04e-1 | 3.06e-1 | 5.91e+1 | 2.42e+6 |
| Test rel. $l^2$ | 3.08e-1 | 2.76e-1 | 2.66e-1 | 3.02e-1 | 2.76e-1 | 2.76e-1 | 1.4 | 1.65e+2 |

We note that data-agnostic methods perform better when tuned slightly towards the problem. To demonstrate this, we experimented on the same system but without noise. In Table H.38 we list relative $l^2$ errors for RFF; the parameters $\sigma_1^{\mathrm{RFF}}$ and $\sigma_2^{\mathrm{RFF}}$ are node/edge encoder and message encoder RFF parameters, respectively. We note better approximations with larger sigmas (lower standard deviation), similar to what we have observed with ELM. This is an important point, highlighting the value of data-agnostic methods in certain cases, especially when their additional tunable parameters are set appropriately. In our main paper experiments, we mainly chose the best-performing random feature method that requires no extra tuning, enabling fast training while maintaining accuracy comparable to gradient-descent-based approaches. However, in real-world scenarios, this fast training could be leveraged to further tune the hyperparameters of RFF or ELM and select the best-performing configuration.

Table H.38: RFF results are displayed where $\sigma_1^{\mathrm{RFF}}$ is the node and edge encoder RFF parameter, and $\sigma_2^{\mathrm{RFF}}$ is the message encoder RFF parameter. Relative $l^2$ errors are displayed.

| | $\sigma_2^{\mathrm{RFF}} = 0.1$ | $\sigma_2^{\mathrm{RFF}} = 1$ | $\sigma_2^{\mathrm{RFF}} = 10$ |
|---|---|---|---|
| $\sigma_1^{\mathrm{RFF}} = 0.1$ | 1.02 | 1.01 | 1.01 |
| $\sigma_1^{\mathrm{RFF}} = 1$ | 1.03 | 9.4e-1 | 6.52e-1 |
| $\sigma_1^{\mathrm{RFF}} = 10$ | 5.34e-1 | 7.04e-2 | 4.76e-2 |

