# OpenReview forum: "Rapid Training of Hamiltonian Graph Networks Using Random Features"
_ICLR.cc/2026/Conference — ICLR 2026 Poster_

### Official Review · Reviewer_nwfX · 2025-10-24

**Soundness:** 3
**Presentation:** 3
**Contribution:** 2
**Rating:** 8
**Confidence:** 5

**Summary:**

This paper proposes a novel iterative method to train Hamiltonian Graph Neural Networks (GNNs) orders of magnitude faster than standard optimizers. The approach leverages random feature sampling over the small shared node, edge, and message MLPs of the GNN, followed by a least-squares fit to determine the final linear layer that outputs the Hamiltonian. To ensure translation and rotation invariance, the particle positions are processed by aligning the simulation frame of reference to the center of mass and rotating the coordinates to a fixed orthonormal basis. The proposed training algorithm is evaluated on four different graph configurations and compared against multiple standard optimizers, achieving at least two orders of magnitude speedup (even outperforming the second-order L-BFGS optimizer). Furthermore, the method is benchmarked against seven state-of-the-art structure-preserving architectures using publicly available datasets.

**Strengths:**

* The method is tested over Neurips 2022 open source benchmarks.
* The paper proposes an interesting way to enforce rotation-invariances in the system.
* The experiments are varied, exploring N-body systems (chains, regular grids) and molecular interactions in Lennard-Jones systems
* The generalization and rollout results are good. These are the precise experiments needed to show the power of structure-preservation in learning conservative dynamics.

**Weaknesses:**

* Even though the method is tested over a wide variety of systems, they are still small scale toy problems.

**Questions:**

* Section 4.3: Is there a particular reason why coosing Hamiltonian GNNs instead of, say, Lagrangian GNNs? Have the authors tried to train the tested architectures (GNODE, LGNN, FGNN, etc) with the presented method to see which architecture performs better?
* Have the authors tried to use simpler invariance tricks, like using relative distances or relative angles?
* Line 1079: $ 5^{-3}$ might refer to $5\cdot 10^{-3}$?
* Line 1079: Here is specified that the molecular dynamics experiments are only rolled-out to 50 timesteps. However, Table 2 shows T=99999 timesteps. Which is the correct one? 50 timesteps is very few time horizon for a molecular system, given that the timestep is very small for stability.
* Equation 6: Is the least squares minimization well-posed? Have the authors found any problems when training? I'm thinking about a bad conditioning number for the normal equations $Z^TZ$, or some degeneracies induced by the rotation-translation invariance.

**Details Of Ethics Concerns:**

I have no ethics concerns.

---

> ### Author Response · Authors · 2025-11-20
>
> We thank the reviewer for their time and consideration. We carefully considered all your concerns, made the necessary improvements to the manuscript, and attempted to address your questions (Q) and comments on weaknesses (W) below.
>
> # W1: Toy Examples
> We acknowledge that we have used only toy examples in our experiments. However, this has been the norm in the Hamiltonian neural network literature to study the new architectural choices \[1, 2, 3, 4, 5\]. Nevertheless, we believe that **testing our approach on larger-scale real-world examples poses a very interesting future work**, where one can rely on **HPC solutions using iterative linear solvers** if the system of interest (Equation 6) becomes very large. This would open up many interesting software challenges, such as **asynchronous data loading from disk** or **parallelization across multiple nodes**, which we consider a very important practical future work.
>
> # Q1: Hamiltonian Formulation and the Other Approaches (FGNN, GNODE, Lagrangian)
> We chose the Hamiltonian formulation because it offers several advantages over other architectures, including **scalar output, second derivative information, hard constraints to conserve the learned Hamiltonian (i.e., the total energy of the system), and the possibility of using symplectic integrators**. Specifically, the **Lagrangian approach is dual to the Hamiltonian formulation**; therefore, our approach would also work almost as presented, and there is no reason to choose one over the other. **The other architectures that do not hard-constrain explicitly to the physics of the system** (GNODE, FGNN) would likely perform worse, simply because **they do not constrain the training to energy-conserving systems**. Trying the random feature approach for these architectures, however, is indeed an interesting future work.
>
> # Q2: Relative distances and angles
> Yes, **relative angles were used** because of the Gram-Schmidt orthogonalization (please see Appendix B.2), and **relative distances were used** in the chain and lattice examples presented in Section 4. We thank the reviewer for pointing this out; **we have now added the relative distance information** to these datasets in Appendix C Lines 1062-1064.
>
> # Q3: Typo on Line 1151
> Thank you very much! We have now corrected this.
>
> # Q4: Number of time steps and step size in the MD experiments
> Line 1151 specifies how the dataset was created for training. $300$ trajectories are created with $50$ timesteps (a very short integration time), which are snapshot at every $20$th timestep (resulting in $3$ datapoints per trajectory). **During testing of a trajectory with an unseen initial condition, we use a very small time-step size of $10^{-5}$ with $10^{5}$ steps**; however, this information was missing. Thanks to the reviewer's question, **we have now included this information** in Appendix D, Table D.18.
>
> # Q5: Is the Least-Squares Well-Posed
> There is no guarantee that the system is well-posed; the rectangular matrix has to be solved with a regularization term in practice (in our case *rcond* parameter of the least-squares solver). From our experience, we did not require a high regularization to achieve good results. We have extended Appendix F by adding Appendix Table F.28 (also shown below), which displays the condition number of the least-squares matrix for a range of hidden dimensions ($d_h$) of encoders and the network width ($d_L$). This was done for a system of $8$ particles. Although the condition numbers we observe are not optimal, in practice, they are within an acceptable range.
>
> |        | $d_L$=128   | $d_L$=256   | $d_L$=512   |
> |--------|-----------|-----------|-----------|
> | $d_h$=8  | 4.266e+05 | 8.985e+06 | 2.612e+08 |
> | $d_h$=16 | 1.525e+05 | 1.687e+06 | 3.545e+07 |
> | $d_h$=32 | 8.105e+04 | 1.352e+06 | 2.974e+07 |
> | $d_h$=64 | 1.072e+05 | 9.625e+05 | 1.766e+07 |
>
> ---
>
> \[1\] Samuel Greydanus, Misko Dzamba, and Jason Yosinski. Hamiltonian neural networks. In H. Wallach, H. Larochelle, A. Beygelzimer, F. d'Alché-Buc, E. Fox, and R. Garnett (eds.), Advances in Neural Information Processing Systems, volume 32. Curran Associates, Inc., 2019.
>
> \[2\] Tom Bertalan, Felix Dietrich, Igor Mezic, and Ioannis G. Kevrekidis. On learning Hamiltonian systems from data. Chaos: An Interdisciplinary Journal of Nonlinear Science, 29(12), 2019.
>
> \[3\] Alvaro Sanchez-Gonzalez, Victor Bapst, Kyle Cranmer, and Peter Battaglia. Hamiltonian Graph
> Networks with ODE Integrators, September 2019.
>
> \[4\] Atamert Rahma, Chinmay Datar, and Felix Dietrich. Training Hamiltonian neural networks without backpropagation. In the NeurIPS 2024 Workshop on Machine Learning and the Physical Sciences. NeurIPS 2024, November 2024.
>
> \[5\] Abishek Thangamuthu, Gunjan Kumar, Suresh Bishnoi, Ravinder Bhattoo, NM Krishnan, and Sayan
> Ranu. Unravelling the performance of physics-informed graph neural networks for dynamical
> systems. Advances in Neural Information Processing Systems, 35:3691–3702, 2022.

---

### Official Review · Reviewer_258K · 2025-10-29

**Soundness:** 2
**Presentation:** 2
**Contribution:** 2
**Rating:** 4
**Confidence:** 3

**Summary:**

The paper introduces Random-Feature Hamiltonian Graph Networks, whose hidden layers are fixed random features, with only the final linear readout solved in one least-squares step instead of iterative gradient descent. The model encodes translation/rotation/permutation invariances, reports 100-600x training time speedups over many optimizers on mass-spring and Lennard-Jones systems, and shows zero-shot scaling from tiny to very large graphs without retraining.

**Strengths:**

1. Replacing long iterative GD with a single convex least-squares solve is simple and yields substantial speedups without complicated tuning.
2. The model encodes translation, rotation, and permutation invariances appropriate for N-body dynamics, which improves data efficiency and generalization within a family.
3. Demonstrates training on small systems and inference on much larger ones without retraining.

**Weaknesses:**

1. The core idea is somehow similar to reservoir computing: both avoid training the feature generation module and optimize only the final readout layer.
2. With a single message passing stage and a linear head, it’s unclear how the method scales to long range or multi scale interactions. There is no ablation on stacking multiple RF blocks.
3. The training formulation assumes no external forces and exact energy conservation. Robustness to mild non-conservation (damping, stochasticity) is only lightly tested.
4. Transfer across distinct topologies is under-explored.

**Questions:**

1. What are the key differences between your approach and reservoir computing?
2. What happens with 2-3 random feature message passing blocks versus a single block? How does performance scale with feature width, and do you observe conditioning issues in the least-squares solve as capacity grows?

---

> ### Author Response · Authors · 2025-11-20
>
> We thank the reviewer for their time and consideration. We carefully considered all your concerns, made the necessary improvements to the manuscript, and attempted to address your questions (Q) and comments on weaknesses (W) below.
>
> # W1/Q1: Differences to Reservoir Computing
> The primary difference is that, unlike our approach, **reservoir computing is not set up to work with graph structured data**, and although reservoir computing variants for graphs exist, these have been focused on graph classification tasks (see the presented results in \[1, 2\]) or regression for non-timeseries prediction (see experiments 4.1.1 and 4.1.2 in \[3\]). Secondly there is a **necessity to tune a large set of hyperparameters for reservoir models** (some important hyperparameters for reservoirs are: input scaling, input connectivity, leak rate, spectral radius of reservoir matrix, connectivity of reservoir matrix); for many problems the hyperparameter tuning is crucial; for instance, the spectral radius of the reservoir matrix $W$ plays an important role in the stability of the reservoir, thus can lead to unstable behavior if not tuned properly. **Our approach significantly alleviates the additional overhead of extensive hyperparameter tuning**.
>
> # W2/Q2: Stacking Multiple Message Passing Blocks
> We thank the reviewer for this interesting question. **An ablation on feature width was already done** (see Appendix Figure F.7). Furthermore, **we have extended Appendix F by adding Appendix Table F.28 (see also below) where we show the condition number** of the least-squares matrix for a range of hidden dimensions ($d_h$) of encoders and the network width ($d_L$), this was done for a system of $8$ particles. In our experience, the *rcond* parameter of the least-squares solver did not need to be set high (\~$10^{-10}$ in most experiments), indicating that the system does not suffer from ill conditioning significantly. This might also be attributed to the physical-loss term introduced in the linear system, Equation 6.
>
> |        |  $d_L$=128  |  $d_L$=256  |  $d_L$=512  |
> |:------:|:---------:|:---------:|:---------:|
> | $d_h$=8  | 4.266e+05 | 8.985e+06 | 2.612e+08 |
> | $d_h$=16 | 1.525e+05 | 1.687e+06 | 3.545e+07 |
> | $d_h$=32 | 8.105e+04 | 1.352e+06 | 2.974e+07 |
> | $d_h$=64 | 1.072e+05 | 9.625e+05 | 1.766e+07 |
>
> While the condition numbers we observe are not optimal and increase with the network width ($d_L$), they remain within an acceptable range in practice.
>
> Thanks to the reviewer's question, we have now also performed an ablation on the number of message passing blocks (and added Appendix Table F.29), again for the same system of $8$ particles, with fixed $d_h=64$ and $d_L=512$. Below we list the results (Test MSE) where the first row uses *summing* as the local pooling and the second row uses *averaging*, and #msg is the number of message passings.
>
> |         |  #msg=1   |  #msg=2   |  #msg=3   |  #msg=4   |  #msg=5   |  #msg=6   |
> |:-------:|:---------:|:---------:|:---------:|:---------:|:---------:|:---------:|
> | summing | 7.713e-06 | 1.866e-05 | 1.271e-04 | 1.150e-03 | 7.620e-04 | 2.222e-03 |
> | average | 2.020e-02 | 1.265e-02 | 1.280e-02 | 1.057e-02 | 1.175e-02 | 1.089e-02 |
>
> We observed that additional message-passing blocks can be beneficial for this mass-spring system when averaging is used. We believe that summing works better than averaging because it implicitly encodes the node degree information by aggregating the neighborhood messages. Each neighborhood message is an output of a softplus activation function and has non-negative values.
>
> Thanks to the reviewer's remark and question, **we have also now included these experiments and discussion in the Ablation Study section** of the Appendix (F) in the manuscript.
>
> # W3: Non-conservative Systems
> In this work, **we only considered Hamiltonian systems, which conserve energy**. However, extending our random-feature approach to dissipative systems with different physical modeling (e.g., GENERIC, Port-Hamiltonian) would be a very interesting next step.
>
> # W4: Transferability
> Transfer across distinct topologies **was explored in terms of node degree** (please see Figure 5 and the discussion in the Limitations section) **and dynamic edges** in the molecular dynamics examples, where the graph topology changes dynamically in each time step and chaotically in longer time horizons.
>
> ---
>
> \[1\] Gallicchio, C., & Micheli, A. (2010, July). Graph echo state networks. In The 2010 international joint conference on neural networks (IJCNN) (pp. 1-8). IEEE.
>
> \[2\] Micheli, A., & Tortorella, D. (2022). Discrete-time dynamic graph echo state networks. Neurocomputing, 496, 85-95.
>
> \[3\] Gallicchio, C., & Micheli, A. (2013). Tree Echo State Networks. Neurocomputing, 101, 319–337. https://doi.org/10.1016/j.neucom.2012.08.017

---

### Official Review · Reviewer_5YK6 · 2025-10-31

**Soundness:** 3
**Presentation:** 3
**Contribution:** 4
**Rating:** 8
**Confidence:** 3

**Summary:**

This paper introduces RF-HGN, a method for training Hamiltonian Graph Networks using random feature sampling instead of iterative gradient-descent optimization. The authors demonstrate significant training speedups compared to other optimizers (Adam, LBFGS, etc.) while maintaining competitive accuracy on mass-spring and molecular dynamics systems.

**Strengths:**

The proposed method offers a significant speedup without sacrificing accuracy in multiple systems.

- comprehensive benchmarking
- elegant solution by merging  Hamiltonian Graph Networks (for physics-informed modeling), Random Features (for fast, non-iterative training), and careful construction of physical invariances (translation, rotation, permutation).
- generalizability from small to larger systems

**Weaknesses:**

- the paper could give more background on why random features work
- acknowledge the limitation more prominently and discuss it
- the acceleration is significant, but the claimed 600× speed-up is overstated; compared to the second-best model, it is actually 150×.

**Questions:**

- The method is presented in the context of Hamiltonian systems. How readily can it be applied to other physics-informed graph networks, such as Lagrangian or Port-Hamiltonian networks, or even non-conservative systems? Does the "random features + linear solve" recipe generalize?

-  The method demonstrates impressive zero-shot generalization to larger systems, but does this generalization hold for structurally heterogeneous systems? For instance, if a model is trained on a regular lattice (where all nodes have the same degree), can it accurately predict the dynamics for a system with a mix of node degrees, or for a node with an unexpectedly high degree not seen during training?

---

> ### Author Response · Authors · 2025-11-20
>
> We thank the reviewer for their time and consideration. We carefully considered all your concerns, made the necessary improvements to the manuscript, and attempted to address your questions (Q) and comments on weaknesses (W) below.
>
> # W1: More Background on Why Random Features Work?
> We have now **added a more comprehensive background on random feature methods** in lines: 121-125.
>
> # W2: Prominent Limitations
> **Our approach does not directly generalize to more complex graph network architectures** (e.g., that utilize attention layers). **We extended the discussion now** in the Limitations and Future work by mentioning the difficulties for adapting our method for different architectures (e.g. attention layers), lines 517-518.
>
> # W3: Overstatement of speedup
> Since the Adam optimizer is the most popular one in literature (e.g., see Table 4), we therefore focused on the speed-up in comparison to this optimizer. However, since the LBFGS is a more modern, second-order optimizer, we understand the point and **have made the necessary changes to the manuscript, reporting 150-600x faster results compared to the SOTA optimizers in the abstract**.
>
> # Q1: Does sampling + linear solve generalize
> Yes, for the particle systems we have considered, **the method generalizes to most physics-informed modeling**: Lagrangian, separable Hamiltonian, Port-Hamiltonian, and GENERIC, for example, can all be formulated as a linear system solve problem.
>
> # Q2: Does the method generalize to unseen node degrees
> **No, the method cannot generalize to unseen node degrees** (see Limitations and Future Work Lines: 514-516) and Figure 5, where training on a 2x2 system does not generalize to systems larger than 3x3 due to the **missing node degree 3 during training**.

---

> > ### Comment · Reviewer_5YK6 · 2025-11-22
> > **Comment:**
> >
> > I would like to thank the authors for their responses. I will keep my score.

---

### Official Review · Reviewer_RhM6 · 2025-10-31

**Soundness:** 2
**Presentation:** 3
**Contribution:** 2
**Rating:** 2
**Confidence:** 4

**Summary:**

This paper proposes RF-HGN, a method to accelerate training of Hamiltonian Graph Networks by replacing gradient-based optimization with random feature sampling and least-squares solvers. The approach samples dense layer parameters randomly (using ELM or SWIM) and optimizes only the final linear layer, achieving 100-600× speedups while maintaining comparable accuracy. The method demonstrates zero-shot generalization from small training systems (8 nodes) to large test systems (4096 nodes) across mass-spring, lattice, and molecular dynamics systems.

**Strengths:**

1. Dramatic Practical Speedups: The 100-600× training acceleration is genuinely impressive and could enable new applications in physics
  simulation.
2. Strong Zero-Shot Generalization: Training on 8-node systems and testing on 4096-node systems demonstrates remarkable scalability - this is perhaps the paper's most valuable contribution.
3. Comprehensive Experimental Validation:
    - Comparison against 15 different optimizers provides robust baselines
    - Multiple physical systems (springs, lattices, molecular dynamics)
    - Use of established NeurIPS 2022 benchmark dataset
4. Physical Consistency: The method maintains energy conservation and incorporates essential symmetries (translation, rotation, permutation invariance).
5. Clear Algorithmic Contribution: The two-stage training procedure (random sampling + least squares) is well-defined and reproducible.

**Weaknesses:**

1. Weak Theoretical Foundation:
    - No convergence guarantees or approximation bounds
    - Limited analysis of when/why the method works
    - Missing connection to random feature theory for this specific setting
2. Poorly Motivated Architectural Choices:
    - No compelling justification for choosing HNNs over alternatives (Lagrangian, Port-Hamiltonian, etc.)
    - Graph networks not well-motivated for many test systems (regular lattices better suited for CNNs)
    - Missing literature review of Hamilton graph neural networks
3. Limited System Complexity:
    - Mostly simple spring-mass systems and basic molecular dynamics
    - ~10% relative error on Lennard-Jones systems suggests limitations for complex potentials
    - No testing on truly challenging physics (e.g., turbulence, phase transitions)
4. Accuracy Trade-offs Not Well Characterized:
    - Sometimes less accurate than second-order methods (LBFGS)
    - Large variance in results (Table 1) raises questions about reliability
    - No principled way to predict accuracy vs. speed trade-offs
5. Scalability Questions:
    - Memory complexity O(MNe) may become prohibitive for very large systems
    - Linear solver bottleneck O(Kd²L) not thoroughly analyzed
    - Integration constant handling appears ad-hoc
6. Missing Key Comparisons:
    - No comparison with other physics-informed ML acceleration techniques
    - No baseline comparisons with structure-specific alternatives (CNNs for lattices)
    - Limited comparison with other random feature applications to physics

**Questions:**

1. Can you provide convergence guarantees or approximation bounds for the random feature approach in the
  physics-informed setting?
2. Why specifically Hamiltonian neural networks? How does performance compare when applying random features to
  Lagrangian or other physics-informed architectures?
3.Can you provide principled guidelines for when accuracy degradation becomes significant? What physical properties are most affected?
4. What are the practical limits of your approach? At what system size does the linear solver become prohibitive?
5. How does the method perform on more challenging physical systems beyond simple spring-mass dynamics?

---

> ### Author Response · Authors · 2025-11-20
>
> We thank the reviewer for their time and consideration. We carefully considered all your concerns and attempted to answer your questions (Q) and comments on weaknesses (W) below.
>
> # W1/Q1: Weak Theory
> **There are convergence guarantees for the sampling approach in the supervised setting** (see \[1\] for SWIM and \[2\] for ELM); however, it does not easily generalize to the graph and physics-informed setting that we treat here. We provide an algorithm, along with extensive computational experiments, and are **currently working on establishing approximation bounds**.
>
> # W2/Q2: Architectural Choice / Missing Literature on Hamiltonian Graph Neural Networks
> **The core of the work is to extend random feature methods to graph networks and demonstrate their applicability in a physics-constrained setting**. The extension to Lagrangian and even to Port-Hamiltonian or GENERIC settings is undoubtedly an interesting area for future work; however, it would not be the primary focus of this work. We updated the manuscript to include one sentence **elaborating more on the most relevant Hamiltonian graph networks in the literature review of graph networks for physics that we are aware of**. If there is any specific literature that we are not aware of, we kindly request the reviewer to specify such relevant work so that we can further improve the manuscript.
>
> # W4/W5/Q3: Accuracy Degradation and Practical Limitations
>
> ## Accuracy Degradation / Principled Guidelines
> Based on our experiments, accuracy **degradation happens when long-term dynamics are predicted in an autoregressive manner, which is due to error accumulation. As a result, the energy and state predictions are affected**. If the **type of neighborhoods for each particle in the test set differs significantly from the type of neighborhood in the training set, then the test performance will be bad**. This is a general behaviour of machine learning methods where the test distribution differs from the training distribution.
>
> ## High Variance
> From Table 1 the variance of our proposed method RF-HGN is way lower than the variance of the SOTA optimizers Rprop, RMSprop, Adam, AdamW, Radam, and Nadam. We hope that these results can address the reliability of our approach in terms of test error variance.
>
> ## Scalability and Practical Limitations of our Approach
> Additionally, one of the most significant practical limits for inference is the **gradient computation of the network due to the inherent Hamiltonian architecture and the associated memory cost** (256GB, as shown in Appendix Table E.26) for very large particle systems (larger than 10_000). Similarly, **for training, the gradient computation of the network and the associated linear system (Equation 6) is again the limiting factor**. In our case, the largest linear system of size (( $(1000 \times 16 \times  6)$ , 512 )) was constructed on a 66GB workstation, and for larger matrices, one would need to rely on HPC solutions. Additionally, please note that no special care is taken regarding memory allocations, and we believe this can be improved further.
>
> # W3/Q4: More Challenging Systems than Mass-Spring
> We consider the **Lennard-Jones potential** example to be more challenging (dynamic graphs, more challenging potential to learn, chaotic system, more neighbors per particle) than the mass-spring dynamics, and **this was already included in our experiments**. If the reviewer has a specific, more challenging system in mind, we would be happy to try our approach on the system.
>
> # W6: Missing Key Comparisons
> **We have already benchmarked our approach against other relevant graph network architectures in literature** (7 different architectures, as shown in Table 4, also see Appendix Figure H.22 and Appendix Figure G.30 for an overview of the features of different architectures) using one **NeurIPS 2022 Benchmark Track publication**. We did not compare our results with CNN architectures or other random feature methods for physics because such methods either do not exist (we are not aware of random feature CNN models) or are not best suited for the problems in physics that we consider here (N-Body Hamiltonian systems).
>
> ---
>
> \[1\] Bolager, E. L., Burak, I., Datar, C., Sun, Q., & Dietrich, F. (2023). Sampling weights of deep neural networks. Advances in Neural Information Processing Systems, 36, 63075-63116.
>
> \[2\] Huang, G. B., Zhu, Q. Y., & Siew, C. K. (2006). Extreme learning machine: theory and applications. Neurocomputing, 70(1-3), 489-501.

---

### Author Response · Authors · 2025-12-03
**Summary of the rebuttal**

Dear AC and Reviewers,

We appreciate the time and effort you have dedicated to reviewing our manuscript. Due to the circumstances in the ICLR discussion phase this year, we hope to make the remaining workload more manageable for you, thus we provide a brief summary of the discussions before Nov 27th. In a good-faith spirit we would like to express that the authors were not aware of the OpenReview bug, and never saw the identities of the persons involved in this review.

---

**Reviewer 5YK6** suggested some additional explanations and modified statements in the manuscript, which we agreed with and **edited accordingly. We also provided a discussion on the possibility of extending our approach** to other types of physics-informed modeling.

**Reviewer nwfX** asked why we chose Hamiltonian GNNs over other systems (e.g., GNODE, FGNN, Lagrangian), which we motivate with attractive properties of the Hamiltonian formulation (scalar output, second derivative information, and the possibility of using symplectic integrators). The reviewer further pointed out some errors in our text and some missing explanations, which **we have addressed in the updated manuscript**. We are truly grateful for the clear, concise, and actionable feedback provided by this reviewer.

**Reviewer 258K** suggested additional comparison experiments when multiple message-passing blocks are stacked together and also asked for more ablation studies regarding feature width scaling (which we already had in the original manuscript, see Appendix Figure F.7 and F.27) and the conditioning of the linear system. We **conducted an ablation study for multiple message passing blocks** (see Appendix Table F.29); **this was supplemented by another ablation study of the condition number in the linear layer with increasing network capacity to investigate conditioning issues** (see Appendix Table F.28). The other questions they raised (concerning the difference between our method and reservoir computing, extensions to non-conservative systems, and the transferability of our method) are also addressed in our response.

**Reviewer RhM6** provided rather general comments regarding the lack of theoretical contribution, the possibility of extending the approach to other physical systems, potential scaling difficulties, and further comparisons that could be conducted either with different architectures or by applying the method to various systems. The reviewer may have overlooked a section in the manuscript (see Section 4.4, esp. Table 4; and the related Table G.30 and Figure H.22) where we already included a SOTA physics-informed architecture benchmark against 7 different architectures. To other questions, we could only answer in a general way, either due to the lack of actionable suggestions (e.g., mentioning general missing literature, or limited comparison against random features used in physics which are not specified and we are unfortunately not aware of) or due to diverging points from our main contribution focus (e.g., modeling turbulence as an N-body Hamiltonian system). We further attempted to address all the reviewer's questions by **providing clarifications with a discussion, updating the manuscript, and adding additional results (Section 4.3) with more complex systems**.

---

The initial scores thus were **(8, 8, 4, 2)**. The manuscript is now updated in light of the reviewer feedback, with **changes highlighted in blue**. We hope this summary can assist your re-evaluation process.

Best wishes,
Authors

---

### Meta-Review · Area_Chair_uQ7N · 2026-01-06

**Summary:**

**Paper Summary**
This paper proposes RF-HGN, a method for training Hamiltonian Graph Networks by replacing iterative gradient-descent optimization with random feature sampling combined with least-squares solvers for the final linear layer. The approach achieves 150 600x training speedups compared to standard optimizers while maintaining comparable accuracy on N-body mass spring and molecular dynamics systems. The  training procedure shows strong zero shot generalization capability, which is proven by showing that training on 8 node systems can predict the dynamics of systems with up to 4096 nodes without retraining.

**Rationale**
The paper appears to make meaningful algorithmic contributions despite the lack of theoretical rigor. Given the genuine disagreement, I will recommend acceptance given that multiple reviewers recommended acceptance.

**Reviewer Concerns:**

Reviewer 5YK6:
- Speedup claim overstatement: The authors revised the abstract to account for both lbfgs and adam
- Random Feature Background: The authors expanded this section
- Attention Limitations: The authors expanded this discussion

Reviewer nwfX:
- Typo at line 1079: corrected
- Time step clarification: corrected
- Least-squares conditioning: Authors conducted a new ablation study

Reviewer 258K:
- Multiple message-passing blocks: Authors conducted requested ablation and presented these results in the appendix
- Feature width scaling: Authors clarified this was already in the original manuscript
- Difference from reservoir computing: Authors provided a detailed response
- Non-conservative systems: Authors acknowledged this is limited to hamiltonian systems <- not addressed

Reviewer RhM6:
- Weak Theory: The reviewer points out that while the method is empirically successful in tested cases, the theoretical contributions are weak.
- Limited Complexity: The reviewer points out that these systems are toy models, and that experiments on very simple potentials (eg LJ) lead to significant errors. This was dropped
- Scalability: The reviewer points out that this scales poorly memory wise. This was dropped.
- Missing Baselines: Reviewer pointed out CNNs and other RF methods for PINN systems. This was dropped

**Reviewer Scores:**

- 5YK6, 8-> 8, and indicated as much
- nwfX, 8 -> 8, the identified weaknesses were minor and all addressed fully
- 258K, 4 -> 5, the substantive critiques were addressed which likely bumped the score up a touch but the reviewer seemed to be poking at limited scope
- RhM6, 2->2, While some concerns were partially addressed, none of the responses seemed in any way convincing

---

### Decision · Program_Chairs · 2026-01-26

Accept (Poster)